# ATLAS: Autoformalizing Theorems through Lifting, Augmentation, and Synthesis of Data

**Xiaoyang Liu[1], Kangjie Bao[1], Jiashuo Zhang[1], Yunqi Liu[1], Yu Chen[1],**
**Yuntian Liu[1], Yang Jiao[3,4] [\*], Tao Luo[1,2] [\*]**

[1] School of Mathematical Sciences, Shanghai Jiao Tong University
[2] Institute of Natural Sciences, MOE-LSC, CMA-Shanghai, Shanghai Jiao Tong University
[3] SPEIT, Shanghai Jiao Tong University
[4] JoinTech Co., Ltd
`{jiaoyang2002, luotao41}@sjtu.edu.cn`

## Abstract

Autoformalization, the automatic translation of mathematical content from natural language into machine-verifiable formal languages, has seen significant progress driven by advances in large language models (LLMs). Nonetheless, a primary barrier to further improvements is the limited availability of parallel corpora that map informal mathematical text to its formal counterpart. To address this limitation, we propose ATLAS (**A**utoformalizing **T**heorems through **L**ifting, **A**ugmentation, and **S**ynthesis of Data), a novel data generation framework designed to produce large-scale, high-quality parallel corpora of theorem statements. Distinct from prior approaches, ATLAS begins with a concept repository, accelerates the improvement of the student model through expert iteration combined with knowledge distillation, and introduces two novel augmentation strategies that exploit the structural characteristics of formal languages. Running the proposed ATLAS framework for 10 iterations, we construct an undergraduate-level dataset of 117k theorem statements and develop the ATLAS Translator by fine-tuning Llama3.1-8B-Instruct with LoRA. This model establishes a new state of the art, demonstrating statistically significant improvements over both the Herald Translator and the Kimina-Autoformalizer across all benchmarks ($p < 0.05$, two-sided t-test). Furthermore, we demonstrate that the full-parameter fine-tuning of a stronger base model on the ATLAS dataset leads to superior performance. The datasets, model, and code are available at `https://github.com/XiaoyangLiu-sjtu/ATLAS`.

## 1 Introduction

In modern mathematics, the escalating complexity of proofs, combined with the increasing reliance on computer-assisted arguments, has raised substantial concerns about reliability. Errors in traditional proofs can remain undetected for extended periods, while computer-assisted proofs frequently lack transparency and are difficult to verify manually, thereby raising issues of trust within the mathematical community. For example, the Four Color Theorem's 1879 proof went unchallenged for over a decade before its flaw was discovered. The first computer-assisted proof in 1976 raised concerns due to its unverifiable computations, prompting further debate. Only in 2005 was the proof formally verified using Coq [4]. To address such issues, formal languages like Isabelle [29], HOL Light [10], Coq, and Lean [7] have been developed to rigorously verify the correctness of proofs.

However, writing mathematical content in formal languages requires significant time and effort, as well as a deep familiarity with these languages, making the process highly labor-intensive. This

---

[\*]Corresponding authors: Yang Jiao, Tao Luo

39th Conference on Neural Information Processing Systems (NeurIPS 2025).

highlights the critical importance of autoformalization, which aims to translate theorem statements and proofs from natural language (NL) into their formal language (FL) counterparts [34]. Since the precise formalization of statements can provide valuable training data for automated theorem proving [21], current research primarily focuses on the autoformalization of theorem statements. In this context, recent progress has shown encouraging results, primarily achieved by fine-tuning large language models (LLMs) with parallel corpora of theorem statements. For clarity, we hereafter refer to theorem statements expressed in natural language as **NL statements**, those expressed in formal language as **FL statements**, and their pairs as **parallel statements**. Furthermore, while ATLAS is a general framework for any formal language, this work focuses on Lean 4 [27] as the target language.

To construct parallel statements, previous studies such as MMA [15] and Herald [9] extract FL statements from Mathlib [35] and generate their NL counterparts using LLMs. However, the limited size of Mathlib imposes restrictions on the scale of the resulting datasets. Alternative methods, including Lean Workbook [51] and DeepSeek-Prover [46], attempt to generate FL statements from NL sources obtained via large-scale web scraping. Although this approach greatly alleviates the limitations on dataset size, it requires extensive pre-processing to obtain high-quality, formalizable NL statements, which significantly reduces overall efficiency. Consequently, it is essential to design a more effective method for generating large-scale, high-quality parallel statements.

In this work, we introduce ATLAS, a data generation framework composed of three key components: Data Lifting, Data Synthesis, and Data Augmentation.

- **Data Lifting.** Unlike previous approaches, this work integrates FL statements and NL statements as its starting point. Specifically, mathematical concepts are abstracted and extracted directly from Mathlib to synthesize NL statements. This method not only overcomes scale limitations but also eliminates the need for data pre-processing.

- **Data Synthesis.** Adopting the knowledge distillation [11] paradigm, teacher models guide the student model's learning process, and the Lean compiler is jointly employed to ensure that the generated FL statements are both semantically accurate and syntactically valid. The resulting parallel statements are hereafter referred to as "synthetic data".

- **Data Augmentation.** The synthetic data are further expanded using two techniques: augmentation via proof and contraposition. The core idea is to leverage Lean 4's capability whereby the Infoview provides real-time updates of the current state after each proof step. The additional parallel statements generated are hereafter referred to as "augmented data".

Finally, the synthetic and augmented data are combined to fine-tune the student model. The expert iteration [30, 31] approach is then employed to iteratively execute ATLAS. After 10 iterations, we build an undergraduate-level dataset comprising 117k parallel statements and train the ATLAS Translator. The performance of the ATLAS Translator shows statistically significant improvements over both the Herald Translator and the Kimina-Autoformalizer across all benchmarks.

Our main contributions are as follows:

1. We propose ATLAS, a novel framework for generating large-scale, high-quality parallel statements. Unlike previous work that starts from NL statements or FL statements, our innovative approach begins by extracting concepts directly from Mathlib through a process we call Data Lifting. Based on these concepts, we employ Data Synthesis and Data Augmentation to synthesize and augment the parallel statements.

2. We introduce the ATLAS dataset and the MathQual dataset. The former comprises 117k undergraduate-level parallel statements, making it one of the largest available. In contrast, the latter contains 465 graduate-level natural language statements, designed to assess the model's autoformalization capability on more challenging data.

3. We develop the ATLAS Translator, which establishes a new state of the art by demonstrating statistically significant improvements over strong baselines across all benchmarks ($p < 0.05$, two-sided t-test). Furthermore, we demonstrate that the full-parameter fine-tuning of a stronger base model on the ATLAS dataset leads to superior performance.

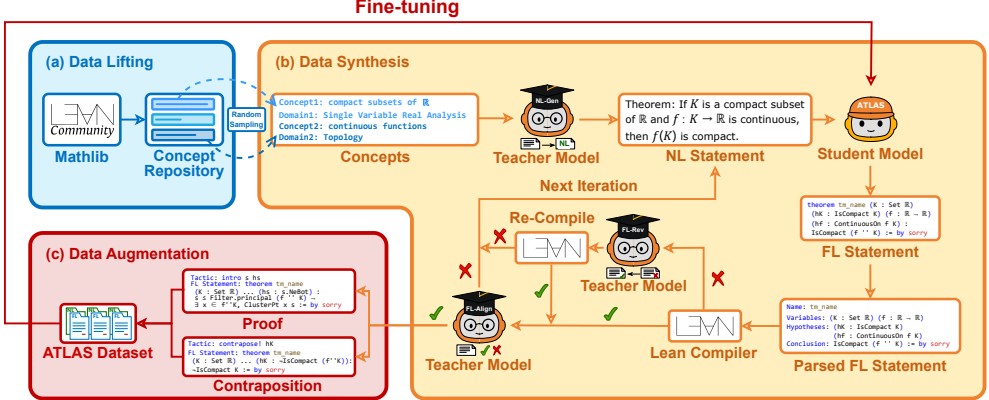

Figure 1: The overview of the proposed ATLAS framework.

## 2 Related Work

**Autoformalization.** The task of autoformalization can be seen as a machine translation problem [39], aiming to convert natural language content into expressions consistent with the target formal language's syntax and vocabulary. Early approaches [6, 40] have utilized neural machine translation techniques to address the autoformalization of theorem statements. With the rapid advancement of LLMs, recent research on LLM-based autoformalization can be broadly categorized into three main paradigms. First, researchers [1, 43, 58] have explored few-shot prompting to enable LLMs to perform autoformalization effectively. Second, some methods [2, 25, 26] further enhance performance by fine-tuning LLMs with parallel statements. Finally, retrieval-augmented generation techniques have been combined with LLMs [54] to achieve additional improvements.

Meanwhile, another line of research focuses on the autoformalization of proofs [16, 55], a more challenging task that closely resembles a simplified form of automated theorem proving [3, 19, 32, 38, 44, 48]. For example, DSP [16] leverages LLMs to generate informal proofs, which are subsequently mapped to formal proof sketches. These sketches then serve as guidance for automated theorem provers to fill in the remaining proof gaps.

**Dataset Generation.** Obtaining large-scale, high-quality parallel corpora of theorem statements remains a significant challenge. Previous efforts [9, 15, 45] have tackled this problem by extracting FL statements from relevant repositories (e.g., Mathlib) and using LLMs to generate their NL counterparts. However, the limited size of these repositories constrains the scalability of the resulting datasets. On the other hand, some approaches [18, 46, 51] take the opposite direction by collecting NL statements from large-scale web sources and translating them into FL representations. While this strategy enables the creation of large-scale datasets, these web-based pipelines rely on extensive preprocessing to filter high-quality, formalizable NL statements, thereby diminishing overall efficiency.

## 3 Methodology

Our framework ATLAS, as illustrated in Figure 1, comprises three components: Data Lifting, Data Synthesis, and Data Augmentation. The framework begins with data lifting, which constructs the concept repository, as described in Section 3.1. Building on this foundation, Section 3.2 details the subsequent data synthesis workflow. Finally, Section 3.3 explains our approach to data augmentation.

### 3.1 Data Lifting

**Mathlib.** Mathlib [35], the most extensive mathematical library within the Lean community, provides a vast collection of formalized notations (e.g., $\|\cdot\|$), concepts, and theorems. This wealth of resources forms the cornerstone of autoformalization. Consequently, importing Mathlib is practically essential before translating a NL statement into a FL statement.

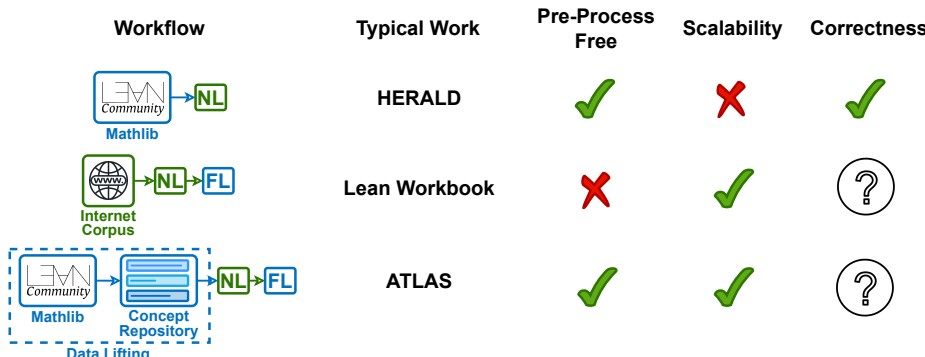

Figure 2: Comparison of different methods for constructing parallel statements.

However, the reliance on Mathlib reveals a critical limitation: when NL statements involve mathematical concepts absent in Mathlib, such as `subgradients`, the autoformalization process is prone to failure. This issue has affected prior work that begins by collecting NL statements. For example, Lean Workbook uses LLMs to categorize 458,692 NL statements and selects 327,870 based on this classification, primarily aiming to exclude NL statements involving concepts not present in Mathlib. In contrast, extracting FL statements from Mathlib does not face this challenge, but the library's size constrains the scalability of the resulting datasets.

In contrast to previous work, our proposed method takes a novel approach by beginning with the extraction of concepts directly from Mathlib through a process we refer to as Data Lifting. By utilizing these concepts, we employ LLMs to synthesize NL statements. As shown in Figure 2, this starting point eliminates the need for pre-processing while ensuring scalability.

**Correctness.** For synthetic data, an important consideration is the correctness of the mathematical propositions themselves. However, in the context of autoformalization, the correctness of the mathematical proposition is secondary, or even insignificant. What is critical in this synthetic process is that the formalized statement is not only syntactically valid but also semantically equivalent to its natural language counterpart. This perspective stems from the primary objective of autoformalization and reflects the realities of mathematical practice, as the correctness of propositions or conjectures is seldom known a priori. Therefore, prior to formal verification, propositions must be precisely formalized in formal language—even if they ultimately turn out to be incorrect.

For example, the following example from the Lean Workbook illustrates that, although the NL statement itself is incorrect (due to the lack of a declaration for the scope of $a, b, c$), the FL statement is semantically equivalent to the NL statement and is syntactically valid. Therefore, this constitutes a valuable piece of synthetic data. We further explore the limitations about correctness in Appendix A.

---

**# lean_workbook_plus_62**

$3(a^2b + b^2c + c^2a) \leq (ab + bc + ca)^2 \leq 9$

```
theorem lean_workbook_plus_62 : ∀ a b c : ℝ, 3 * (a ^ 2 * b + b ^
    2 * c + c ^ 2 * a) ≤ (a * b + b * c + c * a) ^ 2 ∧ (a * b + b *
    c + c * a) ^ 2 ≤ 9 := by sorry
```

---

In the subsequent experimental section, we construct the concept repository based on undergraduate-level mathematical content [2] included in Mathlib. Following Mathlib's organization by domain, topic, and concept, our concept repository comprises 13 domains, 55 topics, and 350 concepts. Further details regarding the composition of the concept repository are provided in Appendix C.

---

[2] https://github.com/leanprover-community/mathlib4/blob/master/docs/undergrad.yaml

## 3.2 Data Synthesis

Upon establishing the concept repository, the module is dedicated to producing a substantial number of high-quality parallel statements by means of knowledge distillation. The following subsections provide a detailed description of each phase, with the specific prompts provided in Appendix F.

**NL Statements Generation.** In this context, the teacher model serves the role of NL Statements Generation (NL-Gen) by randomly sampling concepts from the constructed concept repository to synthesize NL statements. To balance diversity and feasibility, we follow previous approaches [14, 33] by sampling two concepts for each NL statement and the detailed comparisons against MUSTARD [14] are provided in Appendix A. This choice is motivated by the observation that using only a single concept often leads the LLM to generate NL statements that are biased toward well-known, classic mathematical propositions, thus limiting diversity. Conversely, requiring each NL statement to involve many distinct concepts would be overly restrictive, as such problems are uncommon in mathematics and may exceed the capabilities of LLMs.

**NL Statements Translation.** For the synthetic NL statements described above, we employ the student model to translate them into the corresponding FL statements, thereby enabling subsequent tests of syntactic validity and semantic accuracy to ensure the quality of the parallel statements.

**FL Statements Parsing.** Before conducting syntactic validity test on these FL statements, we use the tactic `#check` to decompose each FL statement into the following four components: `theorem_name`, `theorem_variables`, `theorem_hypotheses`, and `theorem_conclusion`. The content is then systematically organized line by line, both within and across these components.

Compared to presenting the entire content on a single line, this line-by-line configuration, especially for statements with substantial content, significantly enhances the clarity of compiler feedback. In particular, error locations become much more explicit, enabling LLMs to more effectively identify and correct errors in FL statements that fail to compile.

**FL Statements Compilation.** In this phase, the Lean compiler is employed to verify the syntactic validity of FL statements by determining whether they can be compiled successfully. If compilation fails, detailed error messages are returned, specifying the location and cause of the error, thereby enabling more efficient subsequent revision. In addition, as the student model is unable to generate headers, a standard header, `import Mathlib`, is automatically appended prior to compilation.

**FL Statements Revision.** For FL statements that fail to compile, we utilize the corresponding NL statements and compilation error messages as context, providing this information to the teacher model serving as FL Statements Revision (FL-Rev) for modification. The modified FL statements are then subjected to a second round of compilation.

Unlike conventional knowledge distillation, the teacher model in our approach is tasked with revising the FL statements generated by the student model based on the provided error messages, rather than generating FL statements directly. There are two primary reasons for this design. First, it is generally much easier to revise existing FL statements than to construct them anew; consequently, this strategy is more likely to produce syntactically valid FL statements after modification. Second, as the performance of the student model improves through iterative learning, the need for FL-Rev diminishes, thus maintaining the efficiency of the overall framework.

**FL Statements Alignment.** For FL statements that pass either the first or second compilation, the teacher model acting as the FL Statements Alignment (FL-Align) evaluator assesses their semantic accuracy in translating the corresponding NL statement, ensuring no information is omitted or mistranslated. Specifically, the model assesses each pair of parallel statements and assigns a rating from three categories: good, average, or poor. Pairs rated as good or average are incorporated into the synthetic data, while those rated as poor, along with FL statements that fail both compilations, have only their NL statements preserved for the next iteration.

### 3.3 Data Augmentation

This module augments the synthetic data obtained in Section 3.2 using two innovative methods: proving these FL statements and converting them into their contrapositives, in order to further expand the scale of the resulting dataset. Figure 3 illustrates an example of the data augmentation process.

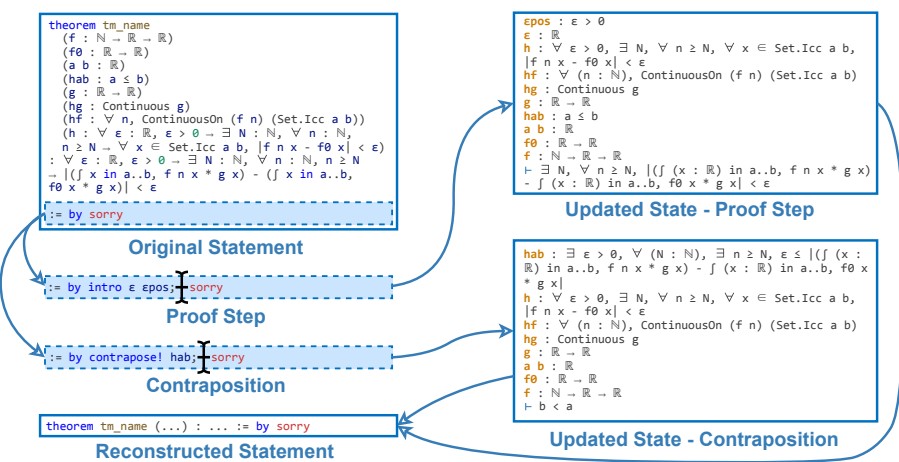

Figure 3: Demonstration of the proof step and contraposition augmentation methods.

**Augmentation via Proof.** For each FL statement in the synthetic data, we use DeepSeek-Prover-V1.5 [47] to generate a corresponding proof. The resulting proof steps are then executed sequentially. Each time a tactic is successfully applied and the proof process is not yet complete, Lean's Infoview updates the proof state, displaying the current variables, hypotheses, and conclusions. Based on this information, new FL statements can be constructed.

**Augmentation via Contraposition.** In Section 3.2 FL Statement Parsing, we obtain the `theorem_hypotheses` for each FL statement. Furthermore, by extracting the names of all hypotheses and applying the `contrapose!` tactic, we can transform the original proposition into an equivalent contrapositive statement for each hypothesis. Leveraging the information provided by Lean's Infoview, new FL statements can again be constructed.

**Augmentation Data Construction.** The aforementioned augmentation operations both rely on Lean Infoview. However, Infoview occasionally results in information loss, particularly concerning type-related details. Although Lean supports implicit type inference, this capability does not extend to all types. To address this, we perform FL Statements Compilation on the augmented FL statements and retain only those that compile successfully.

Furthermore, in the process of data augmentation, the primary consideration is the extent of diversity introduced relative to the original data. To maximize this diversity, we employ the following strategies. For the first augmentation method, we retain only those FL statements produced in the final proof step. For the second method, we utilize the Levenshtein distance [53] to select FL statements that exhibit the greatest dissimilarity from the synthetic FL statements. Finally, we utilize LLMs to translate these augmented FL statements into their corresponding NL statements, thereby constructing parallel statements. The used translation prompt can be found in Appendix F.

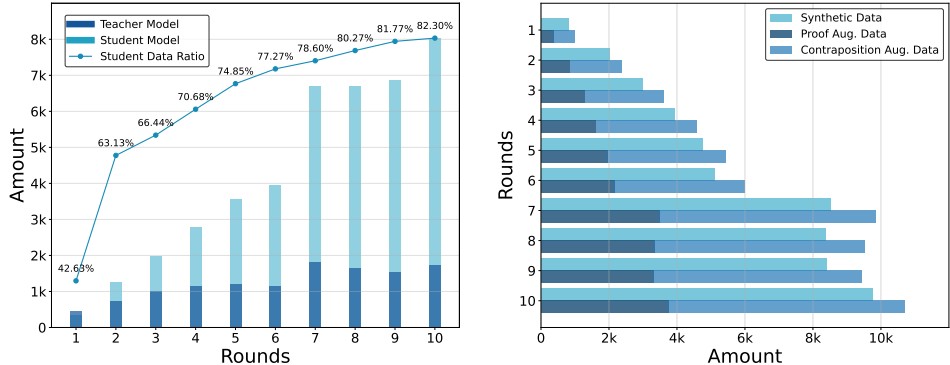

Figure 4: **Data Generation Statistics Across ATLAS Iterations.** Left: The number of synthetic data produced by the teacher and student models at each iteration, with the ratio of student-generated data indicated. Right: The composition of the generated data for each round, including synthetic data, proof augmentation data, and contraposition augmentation data.

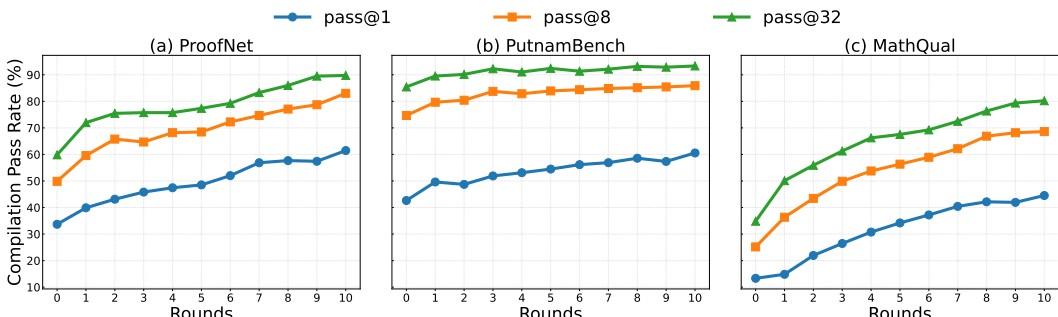

Figure 5: Performance of the student model on benchmarks throughout the iterative training process.

## 4 Experiments

### 4.1 ATLAS dataset Construction

#### 4.1.1 Experimental Setup

We employ DeepSeek-V2.5 [22] as the teacher model and Llama3.1-8B-Instruct [8] as the student model, for which we utilize nucleus sampling [12] with top p=0.9 and a temperature of 0.6 for generation. To demonstrate the effectiveness of our framework, we aim to transform the general-purpose student model into the Lean 4 expert [41].

The training procedure of the student model comprises three stages:

1. **Model Initialization.** We use LeanDojo [50] to extract FL statements from Mathlib and employ LLMs to generate NL statements, constructing a dataset of 56,830 examples. Together with Lean Workbook, this dataset initializes the student model.

2. **Expert Iteration.** In each iteration, 10,000 new NL statements are generated and combined with the remaining NL statements from the previous round to create the synthetic and augmented data. The student model is then fine-tuned on this dataset before proceeding to the next iteration, with this process repeated for a total of 10 rounds.

3. **Final Re-training.** Llama3.1-8B-Instruct is re-trained on the data generated during expert iteration (referred to as the ATLAS dataset) to develop the ATLAS Translator.

All three stages are fine-tuned using LoRA [13] with LLaMA-Factory [57] for 3 epochs, with a total batch size of 128 and a learning rate of 1.0e-5 with a cosine decay schedule. All experiments are conducted on a single NVIDIA A100 GPU with 40GB of memory. In particular, during the Expert

Iteration stage, fine-tuning required only 10 minutes to 1 hour, with minimal hardware requirements and low computational cost, further demonstrating the efficiency of our framework.

### 4.1.2 Experiment Results

As illustrated on the left side of Figure 4, the total amount of synthetic data produced by both the teacher and student models increases across ATLAS iterations. Notably, the proportion of data generated by the student model (indicated by the Student Data Ratio) grows substantially, rising from **42.68%** in round 1 to **82.30%** in round 10. This demonstrates the increasing capacity of the student model to autonomously contribute to the synthetic data as training progresses.

Meanwhile, the right side presents the detailed composition of the generated data for each round. It is evident that with each iteration, not only does the overall quantity of data increase, but the contributions from proof and contraposition augmentation methods also become more prominent. This diversified data augmentation strategy effectively enhances the variety of the dataset, allowing subsequent model iterations to benefit from richer and more diverse training signals. We refer to the synthetic and augmented data as the ATLAS dataset, whose statistics are shown in Table 1.

Figure 5 presents the student model's pass rates on benchmarks across iterative rounds, clearly demonstrating a steady and significant improvement as training progresses. These results indicate that our framework effectively and continuously enhances the student model's autoformalization ability, to a certain extent, since successful compilation is a prerequisite for correct formalization. In addition, Appendix E provides some examples of the synthetic data to illustrate the evolution of the student model's behavior during successive iterations, including cases that achieve successful formalization after several rounds as well as cases that remain unsolved even after 10 iterations.

Table 1: Statistics of the ATLAS dataset

|  | Synthetic Data | Proof Aug. Data | Contraposition Aug. Data | Total |
|---|---|---|---|---|
| ATLAS dataset | 54,641 | 22,103 | 40,401 | 117,145 |

## 4.2 ATLAS Translator Evaluation

### 4.2.1 Experimental Setup

**Dataset.** The datasets used for evaluation are ProofNet [2], PutnamBench [36], and MathQual. The version of ProofNet utilized in this evaluation is sourced from DeepSeek[3]. MathQual is a graduate-level dataset introduced in our work, consisting of 465 NL statements, specifically designed to assess the model's generalization ability on more challenging problems. Detailed information on the construction process and specifics of MathQual can be found in Appendix D.

**Baselines.** We compare ATLAS Translator with the teacher model DeepSeek-V3 [23] (as V2.5 no longer provides API services), the initialization model Llama3.1-Initialization, the previous state-of-the-art model Herald Translator [9], and the latest work Kimina-Autoformalizer [37] to evaluate its performance. DeepSeek-V3 is set with a sampling temperature of 0.7, and the prompt used is provided in Appendix F. Herald Translator and Kimina-Autoformalizer utilize the sampling and prompt settings from their original papers, while Llama3.1-Initialization and ATLAS Translator follow the configurations in Section 4.1.

**Validation Pipeline.** To conduct the evaluation, we follow the validation pipeline described in the Lean Workbook [51] and Herald [9], which includes several key steps:

1. **Translation.** We translate the NL statements into FL statements using the corresponding model.

2. **Compilation.** We use the Lean compiler to verify the syntactic validity of the FL statements.

3. **Back-Translation.** For FL statements that pass compilation, we use InternLM2-Math-Plus-7B [52] to translate them back into NL statements.

---

[3]https://github.com/deepseek-ai/DeepSeek-Prover-V1.5/tree/main/datasets

Table 2: **Overall results of the competing baselines and ATLAS Translator.** The boldface refers to the highest score and the underline indicates the next best result of the models. "**-**" indicates that testing is not performed because the corresponding model uses that dataset during training. "**\***" signifies statistically significant improvements (two-sided t-test with $p < 0.05$) over the best baseline.

| Model | ProofNet | | | PutnamBench | | | MathQual | | |
|---|---|---|---|---|---|---|---|---|---|
| | pass@1 | pass@8 | pass@32 | pass@1 | pass@8 | pass@32 | pass@1 | pass@8 | pass@32 |
| DeepSeek-V3 | 18.82% | 34.07% | 41.35% | 11.53% | 27.74% | 37.33% | 4.90% | 13.42% | 17.29% |
| Llama3.1-Initialization | 23.56% | 42.75% | 51.54% | 19.30% | 45.58% | 62.16% | 6.97% | 15.61% | 22.02% |
| Herald Translator | 31.43% | 64.85% | 78.57% | 20.36% | 52.56% | 71.35% | 10.92% | 31.83% | 45.33% |
| Kimina-Autoformalizer | - | - | - | - | - | - | 19.01% | 38.97% | 50.71% |
| **ATLAS Translator** | **39.46%\*** | **67.28%\*** | **78.71%** | **23.16%\*** | **55.51%\*** | **72.93%\*** | **22.75%\*** | **45.85%\*** | **58.23%\*** |

Table 3: **Ablation study on the three components.** The boldface refers to the highest score and the underline indicates the next best result of the models.

| Model | ProofNet | | | PutnamBench | | | MathQual | | |
|---|---|---|---|---|---|---|---|---|---|
| | pass@1 | pass@8 | pass@32 | pass@1 | pass@8 | pass@32 | pass@1 | pass@8 | pass@32 |
| - *w/o* Synthetic | 24.10% | 51.86% | 67.06% | 15.99% | 42.46% | 58.15% | 11.79% | 29.76% | 40.82% |
| - *w/o* Proof Aug. | 37.41% | 64.85% | 76.39% | 22.88% | 53.23% | 70.71% | **22.97%** | 44.69% | 56.13% |
| - *w/o* Contraposition Aug. | 39.03% | 66.09% | 77.79% | 22.70% | 52.56% | 69.56% | **22.97%** | 44.95% | 57.20% |
| **ATLAS Translator** | **39.46%** | **67.28%** | **78.71%** | **23.16%** | **55.51%** | **72.93%** | 22.75% | **45.85%** | **58.23%** |

4. **NLI Check.** Qwen2.5 [49] is used to compare the back-translated NL statements with the original NL statements to ensure semantic accuracy.

We consider the translation successful if any of these candidates pass both the compilation and NLI check. For a detailed discussion and case study of the validation pipeline's effects, especially the NLI check, please refer to Appendix B.

**Evaluation Metrics.** We use the pass@$k$ metric [5] with $k = 1, 8, 32$, as larger values of $k$ enable LLMs to better realize their potential in generating diverse outputs, which is beneficial for addressing challenging tasks [17, 41]. To reduce randomness, we conduct 5 experiments using the seeds 42, 43, 44, 45, 46, and report the mean of the results. To examine statistical significance, we further perform a two-tailed t-test with $p < 0.05$.

### 4.2.2 Overall Results

The overall results are presented in Table 2. At a glance, we find that the proposed ATLAS Translator outperforms all competing baselines across all datasets and all pass@$k$ metrics, thereby confirming the efficacy of our framework. Further insights will be explored through the subsequent analysis.

**Comparison with Teacher and Initialization Model.** The experimental results clearly demonstrate that ATLAS Translator significantly outperforms its teacher model DeepSeek-V3 and initialization model Llama3.1-Initialization across all benchmarks. Notably, on ProofNet, ATLAS achieves a pass@1 score of 39.46%, nearly doubling DeepSeek-V3's 18.82% and surpassing Llama3.1's 23.56%. More importantly, similar improvements are consistently observed in pass@8 and pass@32 metrics, as well as across other benchmarks, strongly validating the effectiveness of ATLAS framework.

**Comparison with Competing Models.** When examining the comparison with competing models, ATLAS Translator shows remarkable advantages over both the Herald Translator and the newer Kimina-Autoformalizer. With the exception of ProofNet's pass@32, statistically significant improvements are observed in all other metrics and across all other benchmarks. Furthermore, considering that Herald Translator utilizes **1,160k** data points for fine-tuning and that Kimina-Autoformalizer consistently **involves Lean 4 experts** during its training process, it is noteworthy that the ATLAS Translator achieves these results using only **117k** data points and **without any human intervention**. This further underscores the effectiveness of the ATLAS framework.

Table 4: **Additional results of LoRA and full-parameter fine-tuning on various base models with the ATLAS dataset.** "*" denotes full-parameter fine-tuning. Abbreviations: L (Llama-3.1-8B-Instruct), D (DeepSeek-Prover-V1.5-7B-Base), and Q (Qwen2.5-Coder-7B-Instruct). The boldface refers to the highest score and the underline indicates the best result of the baselines. "**-**" indicates that testing is not performed because the corresponding model uses that dataset during training.

| Model | miniF2F | | | ProofNet | | | PutnamBench | | | MathQual | | |
| --- | --- | --- | --- | --- | --- | --- | --- | --- | --- | --- | --- | --- |
| | pass@1 | pass@8 | pass@32 | pass@1 | pass@8 | pass@32 | pass@1 | pass@8 | pass@32 | pass@1 | pass@8 | pass@32 |
| Herald Translator | 76.02% | 93.44% | 95.29% | 31.43% | 64.85% | 78.57% | 20.36% | 52.56% | 71.35% | 10.92% | 31.83% | 45.33% |
| Kimina-Autoformalizer | - | - | - | - | - | - | - | - | - | 19.01% | 38.97% | 50.71% |
| ATLAS Translator (L) | 66.60% | 88.52% | 93.24% | 39.46% | 67.28% | 78.71% | 23.16% | 55.51% | 72.93% | 22.75% | 45.85% | 58.23% |
| ATLAS Translator* (L) | 69.67% | 92.42% | **96.93%** | 47.98% | 74.66% | 86.52% | 38.54% | 73.29% | 84.98% | **40.22%** | 65.81% | 75.48% |
| ATLAS Translator (D) | 67.01% | 90.98% | 96.31% | 39.51% | 69.49% | 81.62% | 25.64% | 59.03% | 75.75% | 24.30% | 49.59% | 63.74% |
| **ATLAS Translator* (D)** | **77.25%** | **93.65%** | 95.90% | **54.99%** | **80.86%** | **88.95%** | **42.49%** | **76.93%** | **87.86%** | 38.92% | **67.31%** | **79.78%** |
| ATLAS Translator (Q) | 66.60% | 91.80% | 96.72% | 38.81% | 71.43% | 84.10% | 29.29% | 66.77% | 82.25% | 28.17% | 54.19% | 68.17% |
| ATLAS Translator* (Q) | 69.88% | 89.75% | 92.62% | 50.67% | 79.51% | 86.25% | 39.91% | 76.93% | 87.56% | 37.63% | 65.59% | 76.34% |

### 4.2.3 Ablation Study

The results of the ablation study are shown in Table 3, where we also conduct 5 experiments using the same seeds and report the mean results. The removal of synthetic data leads to the most significant performance drop across all datasets and metrics, underscoring its critical role in training robustness. In contrast, omitting proof or contraposition augmentation data results in a more moderate decline in performance. Nevertheless, the full model consistently achieves the highest scores or, at the very least, competitive second-best results, thereby validating the synergistic effect of all components.

### 4.2.4 Additional Results

The results of applying LoRA and full-parameter fine-tuning to various base models on the ATLAS dataset are detailed in Table 4. These experiments reveal two key insights.

First, a primary finding is that full-parameter fine-tuning consistently and significantly outperforms the more parameter-efficient LoRA approach. This performance gap is particularly pronounced for the DeepSeek-Prover-V1.5-7B-Base model; on miniF2F [56], full-parameter fine-tuning achieves a pass@1 score of 77.25%, a substantial improvement of over 10 absolute points compared to its LoRA counterpart (67.01%). This trend holds true across all three base models.

Furthermore, the results highlight the critical role of the base model. The DeepSeek-Prover-V1.5-7B-Base model, when fine-tuned on the ATLAS dataset, achieves new state-of-the-art results on most benchmarks, with impressive pass@1 scores of 54.99% on ProofNet and 42.49% on PutnamBench. This outcome is expected, given the model's extensive pre-training on Lean-related corpora. Consequently, we hypothesize that employing a more powerful base model as the student model within our iterative framework would further enhance its overall efficiency.

## 5 Conclusion

In this paper, we propose a novel framework to advance autoformalization by synthesizing and augmenting large-scale, high-quality parallel statements. Our method addresses key limitations of existing approaches, such as the finite amount of data that can be extracted from Mathlib and the extensive pre-processing required for data obtained from web scraping. Through comprehensive experiments, we verify the effectiveness of the ATLAS framework and achieve a new state of the art.

## Acknowledgments and Disclosure of Funding

This work is sponsored by the National Key R&D Program of China Grant No. 2022YFA1008200 (T. L.). We also thank Shanghai Institute for Mathematics and Interdisciplinary Sciences (SIMIS) for their financial support. This research was funded by SIMIS under grant number SIMIS-ID-2025-ST. The authors are grateful for the resources and facilities provided by SIMIS, which were essential for the completion of this work. We appreciate the insightful discussions with Wei Zhao, Xinpu Tu, and Shuyu Yin during the early stages of the project, as well as Tao Zhu's valuable involvement in the human evaluation at a later stage.

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

# A  Motivation, Limitations, and Future Work

**Motivation.**  Our work builds upon MUSTARD [14] by addressing two of its primary limitations. And solutions to these limitations constitute the core contributions of this paper.

- **Data Sourcing.** MUSTARD sources concepts from the Khan Academy Website, which can create a "formalization bottleneck" as some concepts lack a direct counterpart in Mathlib. To overcome this, ATLAS sources concepts exclusively from Mathlib, guaranteeing a valid formalization path for every statement from the start.
- **Generation Efficiency.** MUSTARD's reliance on GPT-4 for correction loops is a costly, low-yield process: over 90% of initial generations fail the prover validation, resulting in a final dataset of only 6k samples. In contrast, the teacher-student distillation framework in ATLAS is highly efficient, dramatically reducing costs and enabling the generation of a much larger dataset of 117k high-quality pairs.

**Limitations.**  A key limitation of our work is that we do not verify the correctness of the synthetic mathematical propositions in the ATLAS dataset. This decision is based on two primary factors. First, the autoformalization task is fundamentally about the fidelity of translation from natural language to formal language; the underlying truth value of a proposition does not alter the core translation challenge. Second, the current capabilities of automated theorem provers are insufficient for reliably verifying a large corpus of undergraduate-level mathematics. For example, the state-of-the-art DeepSeek-Prover-V2-671B [32] only achieves a 7.44% success rate on PutnamBench, making large-scale verification prohibitively challenging and costly.

Nevertheless, we acknowledge that training models on a corpus containing false propositions poses a potential risk for downstream applications, particularly for the automated theorem proving task. To quantify this risk, we performed an analysis to estimate the prevalence of incorrect propositions in our dataset. We randomly sampled 100 propositions from ATLAS and established a ground truth for their correctness using a panel of experts, which included human evaluators alongside advanced LLMs (DeepSeek-R1 and Gemini-2.5-Pro). Using a majority vote consensus from this panel, our analysis classified 60 of the sampled propositions as true and 40 as false. This finding provides a baseline estimate of the dataset's truthfulness and highlights an area for future work in synthetic data refinement.

Another limitation is the diversity of data enhanced by proof. The augmentation-via-proof method is designed to generate novel propositions by treating each step of a formal proof as a semantic transformation. Ideally, the final proposition in a proof chain should be substantially different from the original. However, a limitation of the current implementation is that the practical diversity of these transformations can be constrained, particularly when the proof relies on trivial tactics.

To quantify the textual diversity of augmented data, we conducted an analysis on 100 randomly sampled propositions. We measured the BLEU score between the original and augmented versions, where a lower score signifies greater novelty. The analysis revealed that augmentation-via-proof (Average BLEU: 0.6709) produces less diverse statements on average than augmentation-via-contraposition (Average BLEU: 0.6026). To enhance the former, we plan to implement a stricter filtering mechanism that, for example, disallows augmentations generated from trivial tactics.

**Future Work.**  A primary direction for future work is to implement a more explicit curriculum learning [42] strategy to guide the model's improvement. This strategy comprises two main components:

- **Compositional Complexity.** We will restructure the concept repository into a graph, enabling the systematic generation of a curriculum. New theorems will be synthesized by progressively increasing both the conceptual distance between ideas (i.e., path length in the graph) and the number of concepts required to form a valid statement.
- **Conceptual Hierarchy.** We will introduce a graded concept repository (e.g., high school $\rightarrow$ undergraduate $\rightarrow$ graduate). The model must demonstrate proficiency at one level before unlocking access to the next, creating a structured path toward mastering more advanced topics.

Together, these mechanisms will enable our pipeline to autonomously generate a curriculum of increasing difficulty, creating a powerful virtuous cycle of self-improvement.

A more ambitious goal is to extend our framework from formalizing individual statements to entire mathematical theories. This direction is inspired by dependency-aware, retrieval-based methods like RAutoformalizer [24]. While such methods improve performance by retrieving premises at inference time, our work focuses on data generation. A promising synthesis of these ideas is to integrate the core principle of dependency awareness directly into our data generation process.

Our technical approach for this involves restructuring the concept repository into a dependency-aware tree. During synthesis, we will generate data in a bottom-up, layer-by-layer fashion, explicitly preserving the logical dependencies from foundational axioms to advanced theorems. The resulting structured dataset would be ideal for training next-generation models capable of theory-level autoformalization.

## B    Discussion for Validation Pipeline

In this section, we discuss and examine the reliability and equity of our validation pipeline and evaluate instances of both successful and unsuccessful validations.

**Discussion.**    Recently, there has been some work [20, 24, 28] on automated evaluation of translated FL statements. However, all of these approaches are based on the mutual proof of the translated FL statements and the labeled FL statements. Considering the development of the field of automated theorem proving, it is unrealistic to use this method for automated evaluation on undergraduate and graduate datasets, as the most powerful model currently, DeepSeek-Prover-V2-671B, has only achieved a proof rate of 7.44% on PutnamBench.

Conversely, concerning the validation pipeline, it is important to highlight that validation based on LLMs can sometimes diverge from human judgment, particularly in instances of false positives. Nevertheless, this approach remains standard in the field (e.g., see Lean Workbook [51], Herald [9]) and is currently one of the few viable solutions for large-scale automated evaluation. This is primarily because conducting human expert reviews under the pass@$k$ metric is impractical.

In our implementation of the validation pipeline, especially the NLI check, the primary objective is to establish a fair performance evaluation among different models. As demonstrated in the subsequent experiments, all the baseline models and our proposed model yield comparable results regarding the proportion of false-positive cases, indicating that the validation pipeline is relatively fair.

**Experiment Results.**    To provide a clearer illustration of the false positive cases and the fairness discussed earlier, we conduct an experiment to evaluate the outcomes that pass through the validation pipeline. Specifically, we select the entire ProofNet dataset, consisting of 371 natural language statements, and translate it using the ATLAS Translator, as well as two baseline models: the Herald Translator and the Llama-3.1 Initialization.

The outputs of autoformalization that successfully pass validation are classified by human experts into three distinct categories: correct translations, minor errors, and major errors. These classifications adhere to the evaluation criteria established in Herald [9].

- **Correct Translation.** A correct translation must accurately reflect the mathematical meaning of the natural language statement. In cases where the original statement is ambiguous, we allow the formalized statement to adopt any mathematically valid interpretation.

- **Minor Error.** If the formalized result deviates from the original statement but can be easily corrected by altering a single formal definition or restructuring a single logical relation within the hypothesis or statement, it is classified as a minor error.

- **Major Error.** All other cases are classified as major errors.

As shown in Table 5, the ATLAS Translator and other baseline models also experience false positives, exhibiting comparable false positive rates, which ensures a fair comparison. In Sections B.1, B.2, and B.3, we provide examples for each of these three categories.

Table 5: Human evaluation results of validation passed statements

| Model | Validation Passed | Correct Translation | Minor Error | Major Error |
|---|---|---|---|---|
| Llama3.1-Initialization | 82 | 43 | 30 | 9 |
| Herald Translator | 134 | 79 | 31 | 24 |
| ATLAS Translator | 151 | 99 | 26 | 26 |

## B.1 Correct Examples

**# Example 1**
Suppose that $f$ is holomorphic in an open set $\Omega$. Prove that if $\text{Re}(f)$ is constant, then $f$ is constant.

```
theorem  tm_name  (Ω : Set ℂ) (hΩ : IsOpen Ω) (f : ℂ → ℂ) (hf :
    DifferentiableOn ℂ f Ω) (h : ∀ z ∈ Ω, ∀ w ∈ Ω, (f z).re = (f w)
    .re) : ∀ z ∈ Ω, ∀ w ∈ Ω, f z = f w := by sorry
```

**# Example 2**
Let $X$ be a topological space; let $A$ be a subset of $X$. Suppose that for each $x \in A$ there is an open set $U$ containing $x$ such that $U \subset A$. Show that $A$ is open in $X$.

```
theorem  tm_name {X : Type*} [TopologicalSpace X] (A : Set X) (hA :
    ∀ x ∈ A, ∃ U, IsOpen U ∧ x ∈ U ∧ U ⊆ A) : IsOpen A := by sorry
```

**# Example 3**
Let $p : X \to Y$ be a closed continuous surjective map such that $p^{-1}(\{y\})$ is compact, for each $y \in Y$. (Such a map is called a perfect map.) Show that if $Y$ is compact, then $X$ is compact.

```
theorem  tm_name {X Y : Type*} [TopologicalSpace X] [
    TopologicalSpace Y] [CompactSpace Y] {p : X → Y} (hp :
    Continuous p) (h : Function.Surjective p) (h' : ∀ y : Y,
    IsCompact (p ⁻¹' {y})) : CompactSpace X := by sorry
```

## B.2 Minor Error Examples

**# Example 1**
Show that $\int_0^1 \log(\sin \pi x)dx = -\log 2$.

```
theorem  tm_name (π : ℝ) : ∫ x in (0 : ℝ)..1, Real.log (Real.sin
    (π * x)/π) = -Real.log 2 := by sorry
```

**Analysis:** Incorrectly formalizing $\sin \pi x$ as $\sin(\pi * x)/\pi$.

**# Example 2**
Let $f$ be a real uniformly continuous function on the bounded set $E$ in $R^1$. Prove that $f$ is bounded on $E$.

```
theorem  tm_name (f : ℝ → ℝ) (E : Set ℝ) (hE : IsCompact E ) (hf :
    UniformContinuousOn f E) : ∃ M, ∀ x ∈ E, |f x| ≤ M := by sorry
```

**Analysis:** In the formalization, the condition that the set $E$ is bounded is strengthened to $E$ being compact.

**# Example 3**
Let $R$ be a commutative ring with $1 \neq 0$. Prove that if $a$ is a nilpotent element of $R$ then $1 - ab$ is a unit for all $b \in R$.

```
theorem  tm_name {R : Type*} [CommRing R] [ IsDomain R ] [Nontrivial R
    ] (a b : R) (ha : IsNilpotent a) : IsUnit (1 - a * b) := by
    sorry
```
**Analysis:** In the formalization, an additional assumption was made that $R$ is a domain.

## B.3 Major Error Examples

---

**# Example 1**

If $C_0 + \frac{C_1}{2} + \cdots + \frac{C_{n-1}}{n} + \frac{C_n}{n+1} = 0$, where $C_0, \ldots, C_n$ are real constants, prove that the equation $C_0 + C_1 x + \cdots + C_{n-1} x^{n-1} + C_n x^n = 0$ has at least one real root between 0 and 1.

```
theorem  tm_name (n : ℕ) (hn : 0 < n) (C : Fin (n + 1) → ℝ) (hC : Σ
    i : Fin (n + 1), C i / (i + 1) = 0) : ∃ x : ℝ, 0 ≤ x ∧ x ≤ 1 ∧
    C 0 + C 1 * x + C 2 * x ^ 2 + C 3 * x ^ 3 + C 4 * x ^ 4 + C 5
    * x ^ 5 + C 6 * x ^ 6 + C 7 * x ^ 7 + C 8 * x ^ 8 + C 9 * x ^ 9
    + C 10 * x ^ 10 = 0 := by sorry
```

**Analysis:** The polynomial is written as explicit terms instead of a general sum, losing generality for arbitrary $n$.

**# Example 2**

If $G$ is a group in which $(ab)^i = a^i b^i$ for three consecutive integers $i$, prove that $G$ is abelian.

```
theorem  tm_name {G : Type*} [Group G] [Fintype G] [AddCommMonoid G
    ] [MulAction G G] (a b : G) (i j k : ℕ) (h : i + 1 = j ∧ j + 1
    = k) (h' : (a * b) ^ i = a ^ i * b ^ i) : a * b = b * a := by
    sorry
```

**Analysis:** Omitted the assumptions $(ab)^j = a^j b^j$ and $(ab)^k = a^k b^k$, only stated $(ab)^i = a^i b^i$.

**# Example 3**

Prove that a set $U \subset M$ is open if and only if none of its points are limits of its complement.

```
theorem  tm_name {M : Type*} [MetricSpace M] [TopologicalSpace M] (
    U : Set M) : IsOpen U ↔ ∀ x ∈ U, ∀ y ∈ Uᶜ, x ≠ y := by sorry
```

**Analysis:** This formalization does not faithfully express the conclusion about limit points of the complement."

---

# C  Concept Repository

The complete concept repository is composed of 13 domains, 55 topics, and 350 concepts. For the sake of clarity and brevity, Table 6 only presents a subset of topics from 3 domains, along with their corresponding concepts. And the full repository can be accessed in the open-source material.

# D  MathQual

The MathQual dataset is derived from graduate qualification examinations from multiple universities, including Boston University, Johns Hopkins University, University of Texas at Dallas, University of California, Los Angeles, University of California Riverside, and University of Georgia. Table 7 presents the domains included in the MathQual dataset and the process of creating the dataset is elaborated as follows.

1. Relevant PDF documents are retrieved from official websites.
2. Optical Character Recognition (OCR)[4] technology is employed to convert these documents into Markdown format.
3. High-quality, formalizable problem statements are meticulously selected through a manual filtration process. Notably, for proof problems consisting of multiple sub-questions, we amalgamate the overarching contextual conditions of the main problem with the specific conditions of each sub-question, thereby constructing several distinct problem statements.
4. The problem statements are categorized according to Mathematics Subject Classification (MSC)[5].

---

[4]https://github.com/opendatalab/MinerU
[5]https://zbmath.org/classification

Table 6: Partial list of mathematical concepts in the concept repository

| Domain | Topic | Concept |
|---|---|---|
| Linear Algebra | Fundamentals | vector space, product of vector spaces, vector subspace, quotient space, sum of subspaces, direct sum, complementary subspaces, linear independence, generating sets, bases, existence of bases, linear map, range of a linear map, kernel of a linear map, algebra of endomorphisms of a vector space, general linear group |
| | Duality | dual vector space, dual basis, transpose of a linear map |
| | Finite-dimensional vector spaces | finite-dimensionality, isomorphism with $K^n$, rank of a linear map, rank of a set of vectors, isomorphism with bidual |
| | Multilinearity | multilinear map, determinant of vectors, determinant of endomorphisms, orientation of a $\mathbb{R}$-vector space |
| | Matrices | commutative-ring-valued matrices, field-valued matrices, matrix representation of a linear map, change of basis, rank of a matrix, determinant, invertibility |
| | Endomorphism polynomials | annihilating polynomials, minimal polynomial, characteristic polynomial, Cayley-Hamilton theorem |
| | Structure theory of endomorphisms | eigenvalue, eigenvector, generalized eigenspaces, Jordan-Chevalley-Dunford decomposition |
| Single Variable Complex Analysis | Complex Valued series | radius of convergence, continuity, differentiability with respect to the complex variable, complex exponential, extension of trigonometric functions(cos) to the complex plane, extension of trigonometric functions(sin) to the complex plane, power series expansion of elementary functions(cos), power series expansion of elementary functions(sin) |
| | Functions on one complex variable | holomorphic functions, Cauchy formulas, analyticity of a holomorphic function, principle of isolated zeros, principle of analytic continuation, maximum principle, holomorphic stability under uniform convergence |
| Topology | Topology and Metric Spaces | topology of a metric space, induced topology, finite product of metric spaces, limits of sequences, cluster points, continuous functions, homeomorphisms, compactness in terms of open covers (Borel-Lebesgue), sequential compactness is equivalent to compactness (Bolzano-Weierstrass), connectedness, connected components, path connectedness, Lipschitz functions, uniformly continuous functions, Heine-Cantor theorem, complete metric spaces, contraction mapping theorem |
| | Normed vector spaces on $\mathbb{R}$ and $\mathbb{C}$ | topology on a normed vector space, Banach open mapping theorem, equivalence of norms in finite dimension, norms $\|\cdot\|_p$ on $\mathbb{R}^n$ and $\mathbb{C}^n$, absolutely convergent series in Banach spaces, continuous linear maps, norm of a continuous linear map, uniform convergence norm (sup-norm), normed space of bounded continuous functions, completeness of the space of bounded continuous functions, Heine-Borel theorem (closed bounded subsets are compact in finite dimension), Riesz' lemma (unit-ball characterization of finite dimension), Arzela-Ascoli theorem |
| | Hilbert spaces | Hilbert projection theorem, orthogonal projection onto closed vector subspaces, dual space, Riesz representation theorem, inner product space $l^2$, completeness of $l^2$, inner product space $L^2$, completeness of $L^2$, Hilbert bases, example, the Hilbert basis of trigonometric polynomials, Lax-Milgram theorem |

Table 7: Domain Classification and Problem Counts in MathQual

| Domain | Count |
| --- | --- |
| Algebraic geometry | 2 |
| Algebraic topology | 26 |
| Associative rings and algebras | 11 |
| Calculus of variations and optimal control; optimization | 3 |
| Category theory; homological algebra | 6 |
| Combinatorics | 1 |
| Commutative algebra | 43 |
| Difference and functional equations | 1 |
| Differential geometry | 9 |
| Field theory and polynomials | 32 |
| Functional analysis | 23 |
| Functions of a complex variable | 101 |
| General topology | 24 |
| Global analysis, analysis on manifolds | 18 |
| Group theory and generalizations | 51 |
| Harmonic analysis on Euclidean spaces | 7 |
| Linear and multilinear algebra; matrix theory | 25 |
| Manifolds and cell complexes | 12 |
| Mathematical logic and foundations | 11 |
| Measure and integration | 16 |
| Number theory | 5 |
| Operator theory | 3 |
| Ordinary differential equations | 2 |
| Partial differential equations | 5 |
| Potential theory | 7 |
| Probability theory and stochastic processes | 11 |
| Real functions | 4 |
| Several complex variables and analytic spaces | 3 |
| Topological groups, Lie groups | 3 |
| Total | 465 |

# E Case Study

This section presents both successful and unsuccessful cases of synthetic data during the ATLAS iterative process. Tables 8 to 13 display successful cases generated by ATLAS, while Tables 14 and 15 illustrate the failed cases, each accompanied by corresponding analyses. The FL statements consist of multiple rounds, with content highlighted in red boxes indicating syntactic errors or inconsistencies with the relevant NL statements. Iteration for each data point ceases when there are no syntactic or semantic errors in the FL statement. The cases below demonstrate that, as the model iterates, it is able to formalize mathematical concepts in Lean with increasing accuracy.

Table 8: A successful case generated by ATLAS

**Analysis:** In the first round, the model incorrectly formalized the matrix notation in Lean. However, in the second round, the formalization was correct.

Concept1:  commutative-ring-valued matrices
Domain1:   Linear algebra
Concept2:  determinant
Domain2:   Linear algebra
NL statement:  For any commutative ring $R$ and any two $n \times n$ matrices $A$ and $B$ over $R$, the determinant of their product $AB$ is equal to the product of their determinants, i.e., $\det(AB) = \det(A)\det(B)$ .
FL statements:
Round 1

```
theorem  tm_name {R : Type*} [CommRing R] (n : ℕ)
   (A B : Matrix (Fin n) R))  : (A * B).det = A.det * B.det := by
   sorry
```

Round 2

```
theorem  tm_name {R : Type*} [CommRing R] (n : ℕ) (A B : Matrix (
   Fin n) (Fin n) R) : (A * B).det = A.det * B.det := by sorry
```

Table 9: A successful case generated by ATLAS

**Analysis:** In the first two rounds, the model was unable to correctly formalize the concept of "derivative." However, in the third round, it successfully formalizes this concept, demonstrating that the model effectively masters it after the learning experiences of the first two rounds.

Concept1:  Leibniz formula
Domain1:   Single Variable Real Analysis
Concept2:  hyperbolic trigonometric functions
Domain2:   Single Variable Real Analysis
NL statement:  The derivative of the hyperbolic sine function, $\sinh(x)$, can be expressed using the Leibniz formula as $\sinh'(x) = \cosh(x)$.
FL statements:
Round 1

```
theorem  tm_name (x : ℝ) :  deriv sinh x = cosh x  := by sorry
```

Round 2

```
theorem  tm_name :  deriv sinh = cosh  := by sorry
```

Round 3

```
theorem  tm_name : HasDerivAt (fun x : ℝ => Real.sinh x) (Real.
   cosh x) x := by sorry
```

Table 10: A successful case generated by ATLAS

**Analysis:** In the first round, the model failed to correctly formalize the concepts of "domain" and "being an injection on a certain set." In the second and third rounds, it learned how to formalize "domain," but struggled to accurately formalize "being an injection on a certain set." In the fourth round, it successfully formalized these concepts.

```
Concept1:  metric structure
Domain1:   Single Variable Real Analysis
Concept2:  derivative at a point
Domain2:   Single Variable Real Analysis
```
NL statement: If a function $f$ is differentiable at $c$ and $f'(c) \neq 0$, then there exists a neighborhood $U$ of $c$ such that $f$ is injective on $U$.

FL statements:

Round 1

```
theorem  tm_name (f : ℝ → ℝ) (c : ℝ) (hf : DifferentiableAt ℝ f c
    ∧ deriv f c ≠ 0) : ∃ (U : Set ℝ), (c ∈ U) ∧ (InjectiveOn f U)
    := by sorry
```

Round 2

```
theorem  tm_name (f : ℝ → ℝ) (c : ℝ) (hf : DifferentiableAt ℝ f c)
    (hc : deriv f c ≠ 0) : ∃ U ∈ nhds c, Function.InjectiveOn f U :
    = by sorry
```

Round 3

```
theorem  tm_name (f : ℝ → ℝ) (c : ℝ) (hf : DifferentiableAt ℝ f c)
    (hc : deriv f c ≠ 0) : ∃ U ∈ nhds c,
    Function.Injective (f ∘ (λ x => x) ∘ (U := U) ∘ (λ x => x) ∘ (U := U))
    := by sorry
```

Round 4

```
theorem  tm_name (f : ℝ → ℝ) (c : ℝ) (hf : DifferentiableAt ℝ f c)
    (h'f : deriv f c ≠ 0) :  ∃ U ∈ nhds c, Set.InjOn f U := by
    sorry
```

Table 11: A successful case generated by ATLAS

**Analysis:** In the previous rounds, the model was unable to correctly formalize "the order of a finite group" and the concept of "two coprime positive integers", and there was unnecessary formalization. However, in the eighth round, the formalization was done correctly.

```
Concept1:  direct product of groups
Domain1:   Group Theory
Concept2:  ℤ is a euclidean ring
Domain2:   Ring Theory
```

NL statement: Let $G$ and $H$ be groups. If $G \times H$ is cyclic, then $G$ and $H$ are cyclic and their orders are coprime.

FL statements:

Round 3

```
theorem tm_name {G H : Type*} [Group G] [Group H] (h : IsCyclic (
    G × H)) : IsCyclic G ∧ IsCyclic H ∧
    Nat.coPrime (orderOfOf G) (orderOfOf H) := by sorry
```

Round 4

```
theorem tm_name {G H : Type*} [Group G] [Group H] (hG : IsCyclic
    (G × H)) (hH : IsCyclic (H × G) ) : IsCyclic G ∧ IsCyclic H ∧
    Nat.coPrime (orderOfOf G) (orderOfOf H) := by sorry
```

Round 5

```
theorem tm_name {G H : Type*} [Group G] [Group H] (hG : IsCyclic
    (G × H)) : IsCyclic G ∧ IsCyclic H ∧
    Nat.coPrime (Nat.card G) (Nat.card H) := by sorry
```

Round 6

```
theorem tm_name {G H : Type*} [Group G] [Group H] [Fintype G] [
    Fintype H] (hG : IsCyclic (G × H)) (hH : IsCyclic (H × G) ) :
    IsCyclic G ∧ IsCyclic H ∧
    Nat.coPrime (Fintype.card G) (Fintype.card H) := by sorry
```

Round 7

```
theorem tm_name {G H : Type*} [Group G] [Group H] [Fintype G] [
    Fintype H] (hG : IsCyclic (G × H)) (hH : IsCyclic (H × G) ) :
    IsCyclic G ∧ IsCyclic H ∧ (Fintype.card G).gcd (Fintype.card
    H) = 1 := by sorry
```

Round 8

```
theorem tm_name {G H : Type*} [Group G] [Group H] [Fintype G] [
    Fintype H] (h : IsCyclic (G × H)) : IsCyclic G ∧ IsCyclic H ∧
    (Fintype.card G).gcd (Fintype.card H) = 1 := by sorry
```

Table 12: A successful case generated by ATLAS

**Analysis:** In rounds five to seven, the incorrect formalization "Finset" was used. In round eight, the model revised it to "toFinSet," but it remained incorrect. Finally, in round nine, the formalization was done correctly. Note: The translation results for rounds 1 to 4 were subpar and lacked any significant insights, so they are not displayed.

```
Concept1:  Cauchy-Schwarz inequality
Domain1:   Bilinear and Quadratic Forms Over a Vector Spac
Concept2:  counting measure
Domain2:   Measures and integral calculus
```

NL statement: For any finite set $S$ and functions $f$, $g : S \rightarrow \mathbb{R}$, the Cauchy-Schwarz inequality holds: $\left(\sum_{x \in S} f(x)g(x)\right)^2 \leq \left(\sum_{x \in S} f(x)^2\right)\left(\sum_{x \in S} g(x)^2\right)$.

FL statements:

Round 5 $\sim$ 7

```
theorem  tm_name (S : │FinSet ℝ│) (f g : ℝ → ℝ) : (Σ x in S, f x * g
    x) ^ 2 ≤ (Σ x in S, f x ^ 2) * (Σ x in S, g x ^ 2) := by
    sorry
```

Round 8

```
theorem  tm_name (S : Set ℝ) (hS : S.Finite) (f g : S → ℝ) : (Σ x
    in │S.toFinSet│, f x * g x) ^ 2 ≤ (Σ x in │S.toFinSet│, f x ^ 2) *
    (Σ x in │S.toFinSet│, g x ^ 2) := by sorry
```

Round 9

```
theorem  tm_name {S : Type*} [Fintype S] (f g : S → ℝ) : (Σ x : S,
    f x * g x) ^ 2 ≤ (Σ x : S, f x ^ 2) * (Σ x : S, g x ^ 2) :=
    by sorry
```

Table 13: A successful case generated by ATLAS

**Analysis:** In the first two rounds, there were formalization errors in Lean syntax, while in rounds three and four, the formalization expressed overly strong semantics. However, in round five, the formalization was done correctly.

Concept1:  $K[X]$ is a euclidean ring when $K$ is a field
Domain1:   Ring Theory
Concept2:  inverse trigonometric functions
Domain2:   Single Variable Real Analysis
NL statement:  Let $K$ be a field.  For any polynomial $f(X)$ $\in$ $K[X]$, there exists a polynomial $g(X) \in K[X]$ such that $f(X) \cdot g(X) = 1$ if and only if $f(X)$ is a constant polynomial.  This property is analogous to the invertibility of non-zero constants in the context of inverse trigonometric functions.
FL statements:
Round 1

```
theorem  tm_name {K : Type*} [Field K] (f : Polynomial K) : (∃ (g
    : Polynomial K), Polynomial.mul f g = 1 ) ↔
    f.coeffs.all ((*@λ*@) n => n = 0)  := by sorry
```

Round 2

```
theorem  tm_name {K : Type*} [Field K] (f : K[X]) : (∃ g : K[X], f
    * g = 1) ↔ f = C (1 : K)  := by sorry
```

Round 3

```
theorem  tm_name {K : Type*} [Field K] (f : Polynomial K) : (∃ g :
    Polynomial K, f * g = 1) ↔ f = 1 ∨ f = -1  := by sorry
```

Round 4

```
theorem  tm_name {K : Type*} [Field K] (f : Polynomial K) : (∃ g :
    Polynomial K, f * g = 1) ↔ f = 1  := by sorry
```

Round 5

```
theorem  tm_name {K : Type*} [Field K] (f : Polynomial K) : (∃ g :
    Polynomial K, f * g = 1) ↔ (∃ c : K, f = Polynomial.C c) :=
    by sorry
```

Table 14: A failed case generated by ATLAS

**Analysis:** The model was unable to correctly formalize the concept of a Gaussian integer ring.

```
Concept1:  field ℂ of complex numbers
Domain1:   Ring Theory
Concept2:  greatest common divisor
Domain2:   Ring Theory
```

NL statement: Let $\alpha$ and $\beta$ be non-zero elements in the ring of Gaussian integers $\mathbb{Z}[i]$. Then the greatest common divisor $\gcd(\alpha, \beta)$ in $\mathbb{Z}[i]$ is a Gaussian integer of the form $\gamma = x + yi$, where $x$ and $y$ are integers.

FL statements:

Round 1

```
theorem  tm_name {α : ℂ} (hα : IsAlgebraic ℤ α) (h : IsMonic (
    Polynomial.map (algebraMap ℤ ℂ) (X - C α))) :
    IsEuclideanDomain (ℤ⟨α⟩) := by sorry
```

Round 2

```
theorem  tm_name (α β : ℤ[i]) (hα : α ≠ 0) (hβ : β ≠ 0) : ∃ x y :
    ℤ, ∃ γ : ℤ[i], γ = x + y * I ∧ γ = gcd α β := by sorry
```

Round 3

```
theorem  tm_name (α β : ℤ[i]) (hα : α ≠ 0) (hβ : β ≠ 0) : ∃ γ : ℤ[
    i], γ = gcd α β ∧ ∃ x y : ℤ, γ = x + y * I := by sorry
```

Round 4

```
theorem  tm_name (α β : ℤ[i]) (hα : α ≠ 0) (hβ : β ≠ 0) : ∃ (γ : ℤ
    [i]) (x y : ℤ), γ = x + y * I ∧ γ = gcd α β := by sorry
```

Round 5

```
theorem  tm_name (α β : ℤ[i]) (hα : α ≠ 0) (hβ : β ≠ 0) : ∃ (γ : ℤ
    [i]) (x y : ℤ), γ = x + y * I ∧ γ = gcd α β := by sorry
```

Round 6

```
theorem  tm_name (α β : ℤ[i]) (hα : α ≠ 0) (hβ : β ≠ 0) : ∃ γ : ℤ[
    i], ∃ x y : ℤ, γ = ⟨x, y⟩ ∧ γ = gcd α β := by sorry
```

Round 7

```
theorem  tm_name (α β : ℤ[i]) (hα : α ≠ 0) (hβ : β ≠ 0) : ∃ γ : ℤ[
    i], γ = gcd α β := by sorry
```

Round 8

```
theorem  tm_name (α β : ℤ[i]) (hα : α ≠ 0) (hβ : β ≠ 0) : ∃ γ : ℤ[
    i], γ = gcd α β ∧ ∃ x y : ℤ, γ = x + y * i := by sorry
```

Round 9

```
theorem  tm_name (α β : ℤ[i]) (hα : α ≠ 0) (hβ : β ≠ 0) : ∃ γ : ℤ[
    i], γ = gcd α β := by sorry
```

Round 10

```
theorem  tm_name (α β : ℤ[i]) (hα : α ≠ 0) (hβ : β ≠ 0) : ∃ γ : ℤ[
    i], γ = gcd α β := by sorry
```

Table 15: A failed case generated by ATLAS

**Analysis:** The model was unable to formalize such a relatively complex problem, and there were also certain issues with the natural language statement itself.

Concept1:  Taylor's theorem with Lagrange form for remainder
Domain1:   Single Variable Real Analysis
Concept2:  affine function
Domain2:   Affine and Euclidean Geometry
NL statement:  For a function $f$ that is $n$    $+$    $1$ times differentiable on an interval containing $a$ and $x$, the Taylor polynomial of degree $n$ at $a$ approximates $f(x)$ with an error term that can be expressed as an affine function of $(x-a)^{n+1}$.
FL statements:
Round 1

```
theorem  tm_name (n : ℕ) (f : ℝ → ℝ) (hf : ∀ x, (x ∈ Set.Icc a x)
    → HasFDerivAt (f' x) f x) (x : ℝ) (hx : x ∈ Set.Icc a x) : ∃
    c : ℝ, f x - taylorPolynomial n a f = c * (x - a) ^ (n + 1) :
    = by sorry
```

Round 2

```
theorem  tm_name (a x : ℝ) (n : ℕ) (f : ℝ → ℝ) (hf : ∀ x, f x = Σ
    i in range (n + 1), (f^[i] a * (x - a) ^ i / i!)) : ∃ k : ℝ,
    f x - Σ i in range (n + 1), (f^[i] a * (x - a) ^ i / i!) = k
    * (x - a) ^ (n + 1) := by sorry
```

Round 3

```
theorem  tm_name (n : ℕ) (a x : ℝ) (f : ℝ → ℝ) (hf : ∀ k : ℕ, k ≤
    n → DifferentiableOn ℝ (fun x => iteratedDeriv n f x) (Set.
    Icc a x)) : ∃ c : ℝ, f x - iteratedDeriv n f a = c * (x - a)
    ^ (n + 1) := by sorry
```

Round 4

```
theorem  tm_name (n : ℕ) (a x : ℝ) (f : ℝ → ℝ) (hf : ContDiff ℝ (n
    + 1) f) : ∃ k : ℝ, f x - (f a + Σ i in FinSet.range (n + 1),
    (f^[i + 1] a * (x - a) ^ i / (i + 1)!)) = k * (x - a) ^ (n +
    1) := by sorry
```

Round 5

```
theorem  tm_name (f : ℝ → ℝ) (n : ℕ) (a x : ℝ) (hf : ContDiff ℝ (n
    + 1) f) : ∃ f' g : ℝ, f x = f a + f' * (x - a) + g * (x - a)
    ^ (n + 1) := by sorry
```

Round 6

```
theorem  tm_name (f : ℝ → ℝ) (a x : ℝ) (n : ℕ) (hf : ContDiff ℝ (n
    + 1) f) : ∃ k : ℝ, f x = f a + (f' a) * (x - a) + (1 / (Nat.
    factorial (n + 1))) * (f (n + 1) a) * (x - a) ^ (n + 1) + k *
    (x - a) ^ (n + 1) := by sorry
```

...
Round 10

```
theorem  tm_name (n : ℕ) (a : ℝ) (f : ℝ → ℝ) (hf : ∀ k : ℕ, k ≤ n
    → DifferentiableOn ℝ (fun x => deriv^[k] f x) (Set.Icc a a))
    : ∃ p : ℝ → ℝ, ∀ x : ℝ, |f x - p x| ≤ C * |x - a| ^ (n + 1) :
    = by sorry
```

# F Prompt Templates

This section presents all the prompts used to facilitate the reproducibility of our work.

---

**Prompt Template for NL Statements Translation (DeepSeek-V3)**

Please autoformalize the following problem in Lean 4 with a header. Use the following theorem names: tm_name. Respond strictly with the translation, without any additional explanation.\n\n

---

**Prompt Template for Back-Translation**

[UNUSED_TOKEN_146]user\nConvert the formal statement into natural language:\n"' lean\nformal_statement\n"'[UNUSED_TOKEN_145]\n[UNUSED_TOKEN_146]assistant\n

---

**Prompt Template for NLI Check**

You are an experienced mathematics expert and educator with extensive experience in mathematical problem analysis. I need you to analyze the fundamental nature of the following two mathematical problems.

# Focus on:
1. Core mathematical concepts and principles
2. Problem-solving approaches and methodologies
3. Ultimate objectives of the problems

# Ignore:
1. Variations in wording
2. Changes in contextual scenarios

# Present your answer using exactly this format:
# Analysis\nInsert your analysis here
# Conclusion\nreply ||same|| or ||different|| with "||" format

# Please approach this analysis with professional rigor.
Math Problem 1: {informal_statement}
Math Problem 2: {back_translation}

---

**Prompt Template for NL Statements Generation**

You are an expert mathematics professor tasked with creating proof problems for undergraduate mathematics majors. Your assignment is to construct a proof problem that integrates {concept1} from {domain1} and {concept2} from {domain2}.

Requirements:
1. Create a concise theorem appropriate for undergraduate mathematics majors.
2. The theorem should be brief, not exceeding 50 words.
3. Incorporate both specified concepts into the theorem naturally.
4. State the theorem clearly and concisely.
5. Ensure the theorem is simple enough to be easily translated into Lean4.

Format exactly:
# Answer\nInsert your problem with "||" format, i.e. ||Theorem: Insert the theorem in natural language here.||

## Prompt Template for NL Statements Translation (ATLAS)

You are an expert in the Lean4 theorem prover. Your task is to translate theorems from natural language into formal Lean4 statements. Please follow these guidelines:

1. Carefully analyze the given theorem in natural language.
2. Translate it into a correct and precise Lean4 formal statement.
3. Use the following format for your response: theorem tm_name : The theorem's Lean4 formal statement := by sorry
4. Focus solely on the translation. Do not attempt to prove the theorem or provide additional explanations.
5. Ensure that your translation accurately captures all the mathematical concepts and relationships expressed in the natural language version.
6. Use appropriate Lean4 syntax, including correct use of quantifiers, implications, and mathematical symbols.
7. If the theorem involves specific mathematical structures (e.g., groups, rings, topological spaces), use the corresponding Lean4 definitions and notations.

Remember, the goal is to create a syntactically correct and semantically accurate formalization in Lean4. Your translation should be faithful to the meaning of the original theorem while adhering to Lean4 conventions and best practices.

Now please begin by carefully reading the natural language statement provided, and then proceed with your translation into Lean4.
{informal_statement}

## Prompt Template for FL Statements Revision

You are a math expert and an expert in Lean4. Your task is to modify the Lean4 code based on the given natural language description of a theorem, the corresponding Lean4 code, and the error message from the Lean compiler.

Requirements:
1. Correct the Lean4 code to make it compile successfully.
2. Lean4 code may lack or have additional declarations of certain content. You can add or remove them as much as possible to keep it consistent with the natural language description.
3. No need to import any packages, because Mathlib will be imported by default as import Mathlib.
4. Carefully read the content and provide your modified answer: **Lean4 code**\n{formal _statement}\n**Compiler error messages**\n{compiler_error_messages}\n**natural language statement**\n{informal_statement}

Format exactly:
# Analysis\nInsert your analysis here
# Answer\nInsert your revised Lean4 code with "||" format, i.e. ||theorem tm_name your revised Lean4 code here := by sorry||

## Prompt Template for FL Statements Alignment

You are a math expert and an expert in Lean4. Your task is to check the alignment between the given natural language description of a theorem and the corresponding Lean4 code.

Requirements:
1. Determine whether the Lean4 code is missing declarations of certain entities.
2. Assess whether the Lean4 code accurately represents the theorem described in the natural language.
3. Carefully read the content and provide your answer: **Lean4 code**\n {formal_statement}\n**natural language statement**\n{informal_statement}

Format exactly:
# Analysis:\nInsert your analysis here
# Answer\nreply ||good||, ||average|| or ||poor||

## Prompt Template for FL Statements Translation

You are a math expert and an expert in Lean4. Your task is to translate theorems from Lean4 code into natural language.

Requirements:
1. Focus solely on the translation. Do not attempt to prove the theorem or provide additional explanations.
2. The theorem's natural language statement should be brief, not exceeding 50 words.
3. Carefully analyze the given theorem in Lean4 code {formal_statement} and provide your translation in natural language.

Format exactly:
# Answer\nInsert your translation with "||" format, i.e. ||Theorem: Insert the theorem in natural language here.||

