# OpenReview forum: "ATLAS: Autoformalizing Theorems through Lifting, Augmentation, and Synthesis of Data"
_NeurIPS.cc/2025/Conference — NeurIPS 2025 poster_

### Official Review · Reviewer_eSpp · 2025-06-26

**Clarity:** 3
**Significance:** 2
**Originality:** 2
**Rating:** 4
**Confidence:** 4

**Summary:**

The paper tackles the problem of autoformalization, which is the process of translating informal mathematical statements into equivalent formal statements. The work primarily focuses on the Lean language, but the pipeline is generalizable to other languages. The main contribution of the paper is a concept-repository-based approach to synthetic data generation with augmentation. The authors also present a new autoformalization model, ATLAS, which achieves higher translation accuracy on a variety of datasets.

**Questions:**

Based on the Weaknesses section, here are my questions:

- How can we ensure the originality and sufficient diversity of the augmentation-via-proof method?
- How can we ensure and minimize hallucinations that might occur at various stages of the pipeline?
- How can we guarantee (to some limited degree) the mathematical correctness of the synthetic data?

and a suggestion:
- Please provide some preliminary comparisons with HERALD and the Kimina Autoformalizer on the miniF2F dataset.

**Ethical Concerns:**

["NO or VERY MINOR ethics concerns only"]

**Final Justification:**

After reviewing the rebuttal and responses to other reviewers, I am more positively inclined toward ATLAS. I especially appreciated the experiment on training different model baselines, such inclusions strengthen the work. My recommendations for the authors:

- Update the empirical results section to include the new results
- Expand the discussion of the “correctness” of the translation task; I believe it is a critical point that should be carefully addressed

In light of this, I raise my score to 4.

**Limitations:**

Please review Strengths/Weaknesses section.

**Paper Formatting Concerns:**

None.

**Quality:**

3

**Strengths And Weaknesses:**

## **Strengths**

**Scalable synthetic data generation.** The work introduces two augmentation techniques with minimal inference cost, which contribute to more useful data points in training.

**Concept-based synthetic data.** The work introduces a concept repository that leads to a more diverse set of samples across a variety of topics and, more importantly, their intersections.

**Human-labelling-free fine-tuning.** The pipeline does not involve a human in the loop, which makes the proposed method easier to integrate and scale.

## **Weaknesses**


**Lack of miniF2F comparison.** HERALD translator reaches a very high accuracy of 93.2% on miniF2F, while this work lacks comparison on this dataset. It would be interesting to see the accuracy of ATLAS with a pass128 sampling budget on miniF2F-test/valid.

**High reliance on LMs to generate, curate, correct, and verify the synthetic data.** I understand that the field of formal mathematics is notorious for its lack of quality data, but the pipeline leans heavily on LMs to do the heavy lifting. In this pipeline, I see numerous points of failure where it may generate faulty or misleading data.

During NL statement generation from randomly sampled concepts, LMs may hallucinate, and the final result may omit one of the specified concepts. However, this is a mild weakness that is not a major obstacle, since even incorrect or unsolvable NL statements can be formalized. More importantly, ATLAS uses a teacher model to curate and correct the generated statements. How can we be sure that the teacher model is capable of rewriting and correcting the student model’s mistakes based on compiler feedback, which may also be incomplete (since the compiler stops at the first major error) or misleading (due to the ambiguity of some raised errors)? My main concern is that the teacher model might hallucinate, alter the problem statement, or even trivialize the problem just to pass the compiler check.

**Augmentation via proof.** The authors use prover outputs (such as DSP-v1.5) to augment statements based on the applied tactics. Although this can yield more data through rewriting and the introduction of new hypotheses, it can also clutter the data pool. Provers like DSP-v1.5, as shown in [1], often produce short proofs that rely on built-in solvers, and the applied rewrites are frequently minor or do not substantially affect the NL statements (e.g., $a = b^{-1}$ vs. $a = \frac{1}{b}$ when using the simp tactic). Consequently, this augmentation method leads to high redundancy. It would be interesting to evaluate to what extent this specific technique contributes novel, sufficiently different data points. I believe that augmentation by proof is not as effective as contraposition augmentation.

**No correctness verification of synthetic data.** I understand the authors’ argument that NL- and FL-correctness is secondary in translation tasks and more relevant to formal theorem proving, and that verifying correctness requires proofs, which are scarce and hard to generate. However, since we cannot verify whether these statements are provable, we potentially have many misleading or unprovable statements in the training pool, which the model then uses to learn. How can we ensure some level of correctness in synthetically generated statements?

If we deploy ATLAS in an end-to-end informal-to-formal theorem-proving pipeline, the likelihood of hallucination increases, and the translator may inject wrong or unprovable statements it encountered in the training data. Even though this may not hurt the translator itself (in the context of translation tasks), it introduces risks in the theorem-proving process. As discussed in [2], factual inconsistencies in the dataset increase the likelihood of bias, misinformation, and other artifacts of unverified data. We face hallucinations even with clean, curated datasets; an unverified dataset poses even greater risks for downstream tasks and applications. I recommend deploying more safety guards against producing faulty data (e.g., multiple LLM judges to reduce hallucination or other safeguard methods)

[1] R. Yousefzadeh, X. Cao, A. Ospanov. A lean dataset for international math olympiad: Small steps towards writing math proofs for hard problems. Transactions on Machine Learning Research, 2025.

[2] A. Paullada, I. D. Raji, E. M. Bender, E. Denton, A. Hanna. Data and its (dis)contents: A survey of dataset development and use in machine learning research. Patterns, 2021.

## **Minor typo remarks**

- In line 41, “impose” should be changed to “imposes”.

- In line 504, “translat” should be “translate”.

---

> ### Author Rebuttal · Authors · 2025-07-31
>
> We sincerely thank you for your thorough and detailed review. We appreciate your positive feedback on our scalable, concept-based data generation pipeline and its human-labelling-free design. We have taken your feedback seriously, and in what follows, we provide experimental results and detailed analyses to address each of your concerns.
>
>
> >**Q1: Performance on the miniF2F Benchmark**
>
> **A1:** Thank you for the excellent suggestion to evaluate our model on miniF2F. We agree that this is essential for a thorough comparison to the HERALD Translator baseline. In response, we have now evaluated the ATLAS Translator on it.
>
> **The results show that ATLAS Translator performs comparably to HERALD Translator on miniF2F, and significantly outperforms it on the other three benchmarks.** We also do not report results for Kimina-Autoformalizer as its training set includes the miniF2F benchmark.
>
> |Model|miniF2F|||ProofNet|||Putnam|||MathQual|||
> |-|-|-|-|-|-|-|-|-|-|-|-|-|
> ||pass@1|pass@8|pass@32|pass@1|pass@8|pass@32|pass@1|pass@8|pass@32|pass@1|pass@8|pass@32|
> |HERALD Translator|76.02%|93.44%|95.29%|31.43%|64.85%|78.57%|20.36%|52.56%|71.35%|10.92%|31.83%|45.33%|
> |Kimina-Autoformalizer|-|-|-|-|-|-|-|-|-|19.01%|38.97%|50.71%|
> |ATLAS Translator|77.25%|93.65%|95.90%|54.99%|80.86%|88.95%|42.49%|76.93%|87.86%|38.92%|67.31%|79.78%|
>
> **The ATLAS Translator is based on Deepseek-Prover-V1.5-7B-Base, full-parameter fine-tuned on the ATLAS dataset. Accordingly, all results reported throughout the paper have been updated to reflect this change.*
>
> **The latest HERALD's paper shows that the pass@128 results on minif2f-test and minif2f-valid are 96.7% and 96.3% respectively. Considering that the performance of pass@32 has reached 95.29% and the significance of pass@128 in practical use, we do not test pass@128.*
>
>
> >**Q2: Diversity of Augmentation-via-Proof**
>
> **A2:** Thank you for this insightful question. We agree that the diversity of the augmented data is crucial. In our work (lines 206-208), we **conceptualize each proofstep as a transformation. Ideally, a proposition that has undergone more transformations would be maximally different from the original**. And we take the augmented proposition corresponding to the last proofstep.
>
> However, you correctly point out a key limitation: this theoretical ideal is constrained by the practical capabilities of current automated theorem provers and some trivial tactics. Following your suggestion, we **conduct a new analysis to measure the diversity generated by each strategy**. We randomly sample 100 statements and calculate the BLEU score between the original and augmented versions. In this context, **a lower BLEU score indicates greater textual diversity**. Our analysis yielded the following results:
> - Augmentation-via-Proof: Average BLEU = **0.6709**
> - Augmentation-via-Contraposition: Average BLEU = **0.6026**
>
> These results validate your hypothesis: our current implementation of **Augmentation-via-Proof produces less textually diverse data than Augmentation-via-Contraposition**. This is a valuable finding that provides a clear path for improvement and we will enhance this augmentation method. Specifically, we will implement **stricter filtering by disallowing augmentations generated from trivial proof tactics** (such as simple rewrites with `simp`) and **enforcing a diversity threshold using a metric like Levenshtein distance** to ensure only sufficiently novel statements are added to the training pool.
>
> We thank you for this constructive suggestion. We will incorporate this new analysis and a detailed discussion into the Appendix A to strengthen the paper's discussion of limitations and future work.
>
>
> >**Q3: Concerns about Cheating the Compiler**
>
> **A3:** This is a critical point, and we appreciate you raising it. We agree that the compiler feedback is not always comprehensive and the teacher model may generate hallucinations to cheat the Lean 4 compiler. However, we design our methodology specifically to prevent this type of "reward hacking." To ensure the integrity of our synthetic data, we implement a **two-stage validation process** that all generated data must pass:
>
> 1. **Syntactic Check:** First, the modified formal code is verified for correctness by the Lean 4 compiler. This is the stage where simple cheating could occur.
>
> 2. **Semantic Check**: Second, a powerful verifier model confirms a strict semantic alignment between the compiled formal statement and the original natural language problem.
>
> This dual-check process directly mitigates the risk you've identified. Any attempt by the teacher model to **trivialize or alter the problem to pass the syntactic check would be caught and rejected during the subsequent semantic fidelity check**, as the meaning would no longer match the original intent.
>
>
> >**Q4: Correctness Verification of Synthetic Data**
>
> **A4:** It is important to clarify, as stated in Section 3.1 “Correctness” of the paper, that in the field of Statement Autoformalization, the correctness of the mathematical statements themselves is not the most critical factor. Nevertheless, training models on corpora that contain incorrect statements does pose certain risks for downstream tasks, such as automated theorem proving task. To address this issue, the following **two approaches** can be adopted:
>
> 1. **Judging from NL:** The correctness of the proposition is judged from the perspective of NL through multiple LLMs, and if false, a counterexample should be given. The comprehensive results of multiple LLMs are called **self-consistency methods**, thereby reducing the likelihood of generating incorrect statements.
> 2. **Proving from FL:** Another common practice is to use **automated theorem provers** (such as Deepsee-Prover-V2 and Goedel-Prover-V2) to prove directly from FL. If the statement can be successfully proved, its correctness is thereby ensured.
>
>
> Once again, we thank you for your time and for the insightful feedback that has helped us to significantly strengthen our paper. We hope these new results and clarifications have addressed your concerns, and we respectfully hope for your support.

---

> > ### Comment · Reviewer_eSpp · 2025-08-05
> >
> > I thank the authors for their comprehensive responses to my concerns. In light of the new empirical results and explanations, I will raise my score. I especially appreciated the experiment involving training different base models, which I believe highlights strengths of ATLAS.

---

> > > ### Author Response · Authors · 2025-08-05
> > >
> > > Thank you for your positive feedback and acknowledgment of our efforts in addressing your questions and incorporating revisions. We are pleased to hear about the supplementary experiments on different base models demonstrating the strengths of the ATLAS. We genuinely appreciate your willingness to raise the score for our work.

---

### Official Review · Reviewer_QRN1 · 2025-06-30

**Clarity:** 3
**Significance:** 2
**Originality:** 2
**Rating:** 4
**Confidence:** 4

**Summary:**

This paper presents ATLAS, a novel framework for autoformalization that aims to generate large-scale, qualitative parallel data in both natural and formal mathematical languages. By combining data lifting, synthesis, and augmentation, ATLAS builds a dataset of 117k undergraduate-level statements without manual annotation. The ATLAS translator performs well on multiple benchmarks, marking a meaningful advance in the automation of formal mathematics.

**Questions:**

See Weakness

**Ethical Concerns:**

["NO or VERY MINOR ethics concerns only"]

**Limitations:**

Yes

**Quality:**

3

**Strengths And Weaknesses:**

Strengths:

The paper proposes ATLAS, an autoformalization framework with three core modules. This end-to-end design enables the efficient generation of large-scale parallel corpora in both NL and FL.

By extracting mathematical concepts directly from Mathlib and synthesizing corresponding NL statements, ATLAS avoids reliance on manual annotation or web-based corpus collection.

Weakness:

While ATLAS effectively addresses data scarcity and automates the generation of parallel corpora, its reliance on Mathlib and Lean4 limits its applicability to domains not covered by the library.

The method depends on backbone LLMs for NL synthesis and FL translation. In comparison to LLaMA, more sophisticated large language models, such as those in the Qwen series, could be investigated for potential improvements.

When encountering other out of domain problems, how does the experiment of ATLAS perform? Maybe more analyses are preferred.

---

> ### Author Rebuttal · Authors · 2025-07-31
>
> We sincerely thank you for your positive evaluation of our work, including the end-to-end design of the ATLAS framework for generating large-scale data and its ability to avoid manual annotation. We also wish to thank you for your insightful questions, which have helped us improve the clarity and scope of our paper. In what follows, we address each of the points you raised in detail.
>
>
> >**Q1: Reliance on Lean4/Mathlib**
>
> **A1:** We thank you for raising this important point. We agree that our current implementation is focused on the Lean/Mathlib ecosystem and would like to elaborate on the rationale for this decision and the broader applicability of our methodology.
>
> **Rationale for Mathlib:** Our choice to build upon Mathlib is strategic. As one of the largest and most comprehensive formal mathematics libraries, it provides an ideal foundation for developing and validating a large-scale autoformalization framework. Furthermore, leveraging established libraries is a standard and practical approach in the theorem-proving domain, as seen in prior work such as STP [1] and Lean Workbook [2]. This approach is necessitated by the fact that formalizing novel mathematical domains from scratch remains a formidable challenge for current large language models.
>
> **Generality beyond Lean:** We wish to clarify that the dependency on Lean is a characteristic of our current implementation, not a fundamental limitation of the ATLAS framework. The core principles of our framework—concept lifting, data synthesis, and data augmentation—are designed to be general. **This principled approach could be adapted to other proof assistants, such as Isabelle/HOL or Coq, provided a sufficiently structured library is available.** We will add a discussion to the paper to clarify this distinction and highlight the potential for future generalization.
>
>
> >**Q2: Choice of the Base Model**
>
> **A2:** We appreciate this insightful point. We **select LLaMA as our base model primarily to validate the effectiveness of our framework**. As stated in lines 216-217 of the paper, our objective is to specialize a general-purpose model into a Lean 4 expert. The underlying rationale is that **if our framework can successfully specialize a foundational model, it is expected to yield even more significant performance gains when applied to more advanced LLMs**.
>
> To validate this hypothesis, we conduct a preliminary validation, as comprehensive experiments are not feasible during the brief rebuttal period. We fine-tune several different base models on the ATLAS dataset: Llama3.1-8B-Instruct, DeepSeek-Prover-V1.5-7B-Base, and Qwen2.5-Coder-7B-Instruct, denoted as **ATLAS Translator (L), (D), and (Q)**, respectively—using both LoRA and full-parameter **(denoted by * )** fine-tuning methods. As expected, the results indicate that **stronger models, such as the DeepSeek and Qwen series, achieve significant performance gains over the original LLaMA baseline when fine-tuned on ATLAS dataset**.
>
> |Model|ProofNet|||Putnam|||MathQual|||
> |-|-|-|-|-|-|-|-|-|-|
> ||pass@1|pass@8|pass@32|pass@1|pass@8|pass@32|pass@1|pass@8|pass@32|
> |HERALD Translator|31.43%|64.85%|78.57%|20.36%|52.56%|71.35%|10.92%|31.83%|45.33%|
> |Kimina-Autoformalizer|-|-|-|-|-|-|19.01%|38.97%|50.71%|
> |ATLAS Translator (L)|39.46%|67.28%|78.71%|23.16%|55.51%|72.93%|22.75%|45.85%|58.23%|
> |ATLAS Translator* (L)|47.98%|74.66%|86.52%|38.54%|73.29%|84.98%|40.22%|65.81%|75.48%|
> |ATLAS Translator (D)|39.51%|69.49%|81.62%|25.64%|59.03%|75.75%|24.30%|49.59%|63.74%|
> |ATLAS Translator* (D)|**54.99%**|**80.86%**|**88.95%**|**42.49%**|**76.93%**|**87.86%**|**38.92%**|**67.31%**|**79.78%**|
> |ATLAS Translator (Q)|40.70%|71.43%|84.10%|29.29%|66.77%|82.25%|28.17%|54.19%|68.17%|
> |ATLAS Translator* (Q)|50.67%|79.51%|86.25%|39.91%|76.93%|87.56%|37.63%|65.59%|76.34%|
>
>
> >**Q3: Out-of-Domain (OOD) Performance**
>
> **A3:** This is a valuable question, and we thank you for raising it. To rigorously evaluate our model's generalization capabilities, we've tested it on a challenging out-of-domain benchmark.
>
> Con-NF [3] is an excellent OOD benchmark, as it's based on the Con(NF) library. Con(NF) is a recently published digitization of Randall Holmes’ proof that Quine’s New Foundations is consistent. The benchmark consists of **961 theorems based on a different theoretical basis from our training data source, Mathlib**, making it a true test of generalization. We evaluate our models and existing baselines on this benchmark and report the results below.
>
> - **vs. HERALD Translator:** To evaluate the effectiveness of ATLAS Translator, we first conduct a direct comparison against the **HERALD Translator, a strong baseline also trained on data derived from Mathlib**. Experimental results show that the **ATLAS Translator also performs better than the HERALD Translator**, even in the OOD benchmark.
>
> - **vs. Kimina-Autoformalizer:** Kimina-Autoformalizer achieves a strong **pass@8 score of 40.89%**. This level of performance is achieved by our ATLAS Translator with a **larger pass@32 budget**. We hypothesize that Kimina's strong performance is due to its training corpus, Numina Math 1.5 [4], which is a broad crawl of mathematical knowledge that may include texts on various set theories, including material related to New Foundations. Consequently, **Con-NF may not be a strictly out-of-domain benchmark for Kimina-Autoformalizer**.
>
> Therefore, we argue that ATLAS's performance is a more rigorous test of generalization capability. The ability to achieve a 40% success rate on a demonstrably unseen and conceptually distinct mathematical domain suggests that our framework learns foundational reasoning skills, not just domain-specific patterns. This stricter and more transparent approach to OOD evaluation is a key contribution of our work. A detailed breakdown of this analysis will be included in the Appendix.
>
> |Model|pass@1|pass@8|pass@32|
> |-|-|-|-|
> |HERALD Translator|3.43%|9.05%|14.26%|
> |Kimina-Autoformalizer|**9.37%**|**40.89%**|**68.16%**|
> |ATLAS Translator (L)|7.49%|22.89%|35.07%|
> |ATLAS Translator* (L)|7.28%|21.85%|37.77%|
> |ATLAS Translator (D)|6.04%|19.67%|33.51%|
> |ATLAS Translator* (D)|5.83%|20.40%|33.40%|
> |ATLAS Translator (Q)|7.70%|24.77%|42.98%|
> |ATLAS Translator* (Q)|6.66%|23.93%|37.25%|
>
> We once again thank you for your constructive and insightful feedback. These additions will be fully integrated into the final manuscript. We hope our detailed responses have fully addressed your concerns, and we respectfully hope for your support.
>
> ---
>
> [1] Dong, Kefan, and Tengyu Ma. "STP: Self-play LLM Theorem Provers with Iterative Conjecturing and Proving." Forty-second International Conference on Machine Learning. 2025.
>
> [2] Ying, Huaiyuan, et al. "Lean Workbook: A large-scale Lean problem set formalized from natural language math problems." The Thirty-eight Conference on Neural Information Processing Systems Datasets and Benchmarks Track. 2024.
>
> [3] Liu, Qi, et al. "Rethinking and improving autoformalization: towards a faithful metric and a dependency retrieval-based approach." The Thirteenth International Conference on Learning Representations. 2025.
>
> [4] Li, Jia, et al. "Numinamath: The largest public dataset in ai4maths with 860k pairs of competition math problems and solutions." Hugging Face repository 13.9 (2024): 9.

---

> > ### Comment · Reviewer_QRN1 · 2025-08-05
> >
> > Thank the author for the response, I will keep my score.

---

> > > ### Author Response · Authors · 2025-08-05
> > >
> > > Thank you for your time and for considering our response. We would like to reiterate our sincere gratitude for the recommendations you provided.

---

### Official Review · Reviewer_szrQ · 2025-07-02

**Clarity:** 3
**Significance:** 3
**Originality:** 2
**Rating:** 5
**Confidence:** 3

**Summary:**

This paper presents ATLAS, an autoformalization framework designed to translate natural language mathematical theorems into formal language equivalents. ATLAS establishes its datasets through three key steps: data lifting, data augmentation, and data synthesis. Empirical results demonstrate that the ATLAS translator, which is fine-tuned on the curated ATLAS dataset, significantly outperforms existing baselines and achieves a new state-of-the-art performance.

**Questions:**

- Could the teacher model be further tuned by "exploring more challenging autoformalization problems," potentially allowing the framework to evolve on its own?
- In the experiments, only LoRA was used for fine-tuning the model. Given that the dataset size is relatively sufficient, what is the performance like when employing full-parameter fine-tuning?
- I am also curious about the expert iteration stage within the framework. Specifically, it is possible to synthesize formal theorems based on the collected formal concepts, then transform them into natural language versions, followed by SFT training of the student model instead of the expert iteration. This approach could also be enhanced through data augmentation. What are the results of applying this method?

**Ethical Concerns:**

["NO or VERY MINOR ethics concerns only"]

**Final Justification:**

This paper addresses a critical problem in formal theorem proving. The proposed method and dataset are valuable contributions to the community, and I recommend accepting this paper.

**Limitations:**

Yes

**Quality:**

3

**Strengths And Weaknesses:**

### Strengths
- This paper focuses on a significant problem in the field of formal theorem proving.
- The overall framework presented is both interesting and promising. Moreover, its generality has the potential to inspire further enhancements and applications.
- The dataset created is of high quality, and experimental results indicate that the model fine-tuned on this collection of 117K data points can achieve results comparable to existing baselines, which are typically trained on much larger datasets.

### Weaknesses
- Compared to existing baselines, the ATLAS Translator does not demonstrate significant improvements, particularly on the ProofNet and PutnamBench datasets.
- The ablation study reveals that the primary gains stem from the data synthesis of the teacher models, while the proposed data augmentation methods have somewhat less impact.
- There is some concern regarding the validation pipeline. The existing methods [20, 24, 28] rely solely on automated theorem provers to verify the logical equivalence between formalization results and ground truths. This may be considerably easier than actually proving the theorems themselves (if not, please correct me). Therefore, it would be beneficial to consider incorporating this approach as an additional evaluation method.
- While I agree that the correctness of a theorem may be secondary in certain contexts, many tasks involving LLM training for theorem proving necessitate the formalization of correct and provable theorems as training materials. Thus, it would be beneficial to report some results in this direction on a smaller test set.

---

> ### Author Rebuttal · Authors · 2025-07-31
>
> Thank you for your detailed and insightful review of our paper. We are especially encouraged that you recognize the significance of the problem we are tackling and find our overall framework to be "interesting and promising." We are particularly grateful for your observation that our high-quality dataset allows ATLAS Translator to achieve results comparable to baselines that use much larger training sets, which is a central point of our contribution. We also deeply appreciate your constructive critiques and thoughtful questions and  address each of your questions and comments in detail below.
>
>
> >**Q1: Limited Performance Gains and Full-Parameter Fine-Tuning**
>
> **A1:** Thank you for raising these critical points. To thoroughly investigate, we perform an expanded experiment. This involved fine-tuning three distinct base models on the ATLAS dataset: Llama3.1-8B-Instruct, DeepSeek-Prover-V1.5-7B-Base, and Qwen2.5-Coder-7B-Instruct, denoted as **ATLAS Translator (L), (D), and (Q)**, respectively-using both LoRA and **full-parameter (denoted by * )** fine-tuning methods, and benchmarking them against existing baselines.
>
> **These new results, using full-parameter fine-tuning, resolve the concern about limited performance gains.** Our best model, ATLAS Translator* (D), now establishes a new SOTA across all benchmarks:
> - On **ProofNet**, it boosts pass@1 performance over HERALD by **23 absolute points** (54.99% vs. 31.43%).
> - On **PutnamBench**, it **more than doubles** the pass@1 performance of HERALD (42.49% vs. 20.36%).
> - On **MathQual**, it **more than triples** the pass@1 performance of HERALD (38.92% vs. 10.92%).
>
> Furthermore, the results confirm your intuition: **full-parameter fine-tuning consistently and significantly outperforms LoRA across all three base models**. We have updated the paper to reflect this new SOTA performance.
>
> |Model|ProofNet|||Putnam|||MathQual|||
> |-|-|-|-|-|-|-|-|-|-|
> ||pass@1|pass@8|pass@32|pass@1|pass@8|pass@32|pass@1|pass@8|pass@32|
> |HERALD Translator|31.43%|64.85%|78.57%|20.36%|52.56%|71.35%|10.92%|31.83%|45.33%|
> |Kimina-Autoformalizer|-|-|-|-|-|-|19.01%|38.97%|50.71%|
> |ATLAS Translator (L)|39.46%|67.28%|78.71%|23.16%|55.51%|72.93%|22.75%|45.85%|58.23%|
> |ATLAS Translator* (L)|47.98%|74.66%|86.52%|38.54%|73.29%|84.98%|40.22%|65.81%|75.48%|
> |ATLAS Translator (D)|39.51%|69.49%|81.62%|25.64%|59.03%|75.75%|24.30%|49.59%|63.74%|
> |ATLAS Translator* (D)|**54.99%**|**80.86%**|**88.95%**|**42.49%**|**76.93%**|**87.86%**|**38.92%**|**67.31%**|**79.78%**|
> |ATLAS Translator (Q)|40.70%|71.43%|84.10%|29.29%|66.77%|82.25%|28.17%|54.19%|68.17%|
> |ATLAS Translator* (Q)|50.67%|79.51%|86.25%|39.91%|76.93%|87.56%|37.63%|65.59%|76.34%|
>
>
> >**Q2: Additional Evaluation Method**
>
> **A2:** We sincerely thank you for this excellent suggestion to incorporate a more rigorous evaluation method. The three existing methods you mentioned are roughly introduced as follows:
>
> - The first method [1] is **specific to the Isabelle** theorem prover, relying on built-in tactics like `Sledgehammer`, `by presburger` and so on.
> - The second method [2] uses the Lean theorem prover to verify **bidirectional equivalence between the model's output and the ground truth**. It builds upon the heuristic tactic `exact?` by also employing a LLM to find the required proofs.
> - The third method [3], designed for the Lean theorem prover, is **restricted to the domain of Euclidean Geometry**.
>
> Given the limitations of the other approaches, we adopt the second method (BEq@k, k=1,8) and re-evaluate all models on ProofNet. This results **further corroborate the conclusions from the LLM-as-a-Judge evaluation**, demonstrating that the full-parameter fine-tuned model achieves superior performance over both the LoRA-tuned variant and the HERALD baseline on this more rigorous metric. A detailed breakdown of this analysis will be included in the Appendix.
>
> |Model|BEq@1|BEq@8|
> |-|-|-|
> |HERALD Translator|4.58%|7.55%|
> |Kimina-Autoformalizer|-|-|
> |ATLAS Translator (L)|5.39%|9.16%|
> |ATLAS Translator* (L)|6.47%|9.97%|
> |ATLAS Translator (D)|5.66%|9.16%|
> |ATLAS Translator* (D)|6.74%|9.16%|
> |ATLAS Translator (Q)|5.93%|8.09%|
> |ATLAS Translator* (Q)|7.01%|8.89%|
>
>
> >**Q3: Verification of Theorem Correctness/Provability**
>
> **A3:** Thank you for this excellent suggestion. To address the crucial question of whether the formalized statements are provable, we conduct a preliminary study.
>
> We **randomly sample 100 propositions** from the ATLAS dataset. To establish a ground truth for their provability, we employ a panel of experts, including DeepSeek-R1, Gemini 2.5 Pro, and human evaluators, to collectively assess the truth value of each proposition. This process results in a set of **60 true and 40 false** propositions by **self-consistency strategy**.
>
> Furthermore, we will expand this into a full discussion in the Appendix, rather than merely listing it as a limitation.
>
>
> >**Q4: Potential for Framework Self-Evolution**
>
> **A4:** This is an insightful question regarding the framework's long-term potential for autonomous improvement. We agree that enabling the system to evolve by tackling progressively more difficult problems is a critical direction for future work.
>
> Our **current framework already includes a foundational mechanism** for this: natural-language statements that the model **fails to formalize are re-introduced into the candidate pool for subsequent attempts**. This process implicitly increases the average task difficulty by exposing the model to its previous failures.
>
> Building directly on this principle, we plan to implement a **more explicit curriculum learning strategy for self-evolution** with two primary components:
>
> 1. **Compositional Complexity:** We will structure the concept library as a graph or a tree, which allows us to systematically generate a curriculum of increasing difficulty. New theorems will be synthesized by progressively increasing both the conceptual distance between ideas (e.g., path length in the graph) and the total number of concepts required to form a valid statement.
>
> 2. **Conceptual Hierarchy:** We will introduce a graded concept repository (e.g., high-school → undergraduate → graduate level). The model would need to demonstrate proficiency at one level before unlocking the next, creating a structured path to mastering more advanced topics.
>
> Together, these mechanisms would enable the pipeline to autonomously generate a curriculum of increasing difficulty, creating a powerful virtuous cycle of self-improvement that does not depend on new external data or annotations.
>
>
> > **Q5: Discussion about Alternative Framework Designs**
>
> **A5:** Thank you for this excellent question regarding alternative framework designs. We did, in fact, explore a "formal-first" synthesis approach similar to what you described during our initial investigations. We ultimately chose our current "natural-first" ATLAS framework for two primary, empirically-grounded reasons:
> 1. **High Cost and Low Yield of Formal Synthesis:** Our experiments confirm that directly synthesizing formal statements from concepts is highly inefficient. For instance, when we prompt DeepSeek-V3 to generate 300 formal statements, **only 2.3% (7 out of 300) successfully pass verification by the Lean 4 prover**. This extremely low yield makes the "formal-first" approach difficult and costly to scale.
>
> 2. **LLM Proficiency and Data Scarcity**: This low success rate reflects a broader challenge: current LLMs are generally more proficient at generating fluent, diverse natural language than they are at producing syntactically and logically perfect formal code. This is partly due to the relative scarcity of formal mathematics corpora in their pre-training data compared to the vast amount of natural text. For instance, in the **math-related code corpus AlgebraicStack [4], Lean accounts for only 2.6% (285M/10955M).**
>
> In contrast, our ATLAS framework strategically leverages the strengths of modern LLMs (their powerful natural language capabilities) and then uses the Lean prover as a targeted, efficient verifier. This design has proven to be a far more cost-effective and scalable method for generating the large, high-quality dataset that powers our SOTA results.
>
>
> Once again, we sincerely thank you for your detailed and constructive feedback. Your suggestions have been instrumental in improving our work and we hope these revisions have fully addressed your concerns and questions.
>
> ---
>
> [1] Li, Zenan, et al. "Autoformalize Mathematical Statements by Symbolic Equivalence and Semantic Consistency." The Thirty-eighth Annual Conference on Neural Information Processing Systems. 2024.
>
> [2] Liu, Qi, et al. "Rethinking and improving autoformalization: towards a faithful metric and a dependency retrieval-based approach." The Thirteenth International Conference on Learning Representations. 2025.
>
> [3] Murphy, Logan, et al. "Autoformalizing Euclidean Geometry." Forty-first International Conference on Machine Learning. 2024.
>
> [4] Azerbayev, Zhangir, et al. "Llemma: An Open Language Model for Mathematics." The Twelfth International Conference on Learning Representations. 2024.

---

> > ### Comment · Reviewer_szrQ · 2025-08-07
> >
> > Thank you for the detailed response and additional experiments. All of my concerns are addressed, and I am pleased to increase my score to 5.

---

> > > ### Author Response · Authors · 2025-08-07
> > >
> > > We are very grateful for your detailed feedback and for the time you took to consider our response and new experiments. Thank you so much for your support and for raising your score. Your suggestions have been invaluable in strengthening our paper.

---

### Official Review · Reviewer_z8Zj · 2025-07-15

**Clarity:** 3
**Significance:** 2
**Originality:** 2
**Rating:** 4
**Confidence:** 4

**Summary:**

The paper proposes ATLAS, a framework for autoformalizing mathematical theorems by generating high-quality parallel datasets of natural and formal statements. Starting from Mathlib concepts, it uses data lifting, synthesis, and augmentation (via proofs and contraposition) to build a 117k-statement dataset. The resulting ATLAS Translator outperforms prior models like HERALD and Kimina on benchmarks, achieving new state-of-the-art results with less data and no human supervision.

**Questions:**

- As a core baseline, HERALD benchmarked heavily on MiniF2F. I was wondering what is the performance of ATLAS on it (MiniF2F)?

**Ethical Concerns:**

["NO or VERY MINOR ethics concerns only"]

**Limitations:**

yes

**Paper Formatting Concerns:**

n.a.

**Quality:**

3

**Strengths And Weaknesses:**

Pros
- Well written paper, easy to follow. I especially appreciate the super illustrative examples in the appendix that qualitatively explain the error types.
- Amazing performance with super affordable compute budget (i.e., one A100).

Cons:
- Each of the component does not appear super novel: data lifting (MUSTARD), synthesis (MMA, ProofNet, Lean Workbook), augmentation (DeepSeek-Prover). Nevertheless, the authors did a good job to put everything together.
- Despite briefly mentioned, I believe slightly more comparisons against MUSTARD [1] and RAutoformalizer [2] might be worthy.

[1] Huang, Yinya, et al. "Mustard: Mastering uniform synthesis of theorem and proof data." arXiv preprint arXiv:2402.08957 (2024).

[2] Liu, Qi, et al. "Rethinking and improving autoformalization: towards a faithful metric and a dependency retrieval-based approach." The Thirteenth International Conference on Learning Representations. 2025.

---

> ### Author Rebuttal · Authors · 2025-07-31
>
> We sincerely thank you for your positive feedback on our paper's clarity and impressive performance-to-compute ratio. We also appreciate your constructive questions, which have helped us further strengthen the manuscript. Below, we address the points you raised.
>
>
> > **Q1: Performance on the miniF2F Benchmark**
>
> **A1:** Thank you for the excellent suggestion to evaluate our model on miniF2F. We agree that this is essential for a thorough comparison to the HERALD Translator baseline. In response, we have now evaluated the ATLAS Translator on it.
>
> **The results show that ATLAS Translator performs comparably to HERALD Translator on miniF2F, and significantly outperforms it on the other three benchmarks.** We also do not report results for Kimina-Autoformalizer as its training set includes the miniF2F benchmark.
>
> |Model|miniF2F|||ProofNet|||Putnam|||MathQual|||
> |-|-|-|-|-|-|-|-|-|-|-|-|-|
> ||pass@1|pass@8|pass@32|pass@1|pass@8|pass@32|pass@1|pass@8|pass@32|pass@1|pass@8|pass@32|
> |HERALD Translator|76.02%|93.44%|95.29%|31.43%|64.85%|78.57%|20.36%|52.56%|71.35%|10.92%|31.83%|45.33%|
> |Kimina-Autoformalizer|-|-|-|-|-|-|-|-|-|19.01%|38.97%|50.71%|
> |ATLAS Translator|77.25%|93.65%|95.90%|54.99%|80.86%|88.95%|42.49%|76.93%|87.86%|38.92%|67.31%|79.78%|
>
> **The ATLAS Translator is based on Deepseek-Prover-V1.5-7B-Base, full-parameter fine-tuned on the ATLAS dataset. Accordingly, all results reported throughout the paper have been updated to reflect this change.*
>
>
> > **Q2: Detailed Comparison to MUSTARD and RAutoformalizer**
>
> **A2:** Thank you for highlighting these two important works. Based on your suggestion, we have now added a more detailed discussion in Appendix A: Motivation, Limitation and Future Works.
>
> **MUSTARD:** MUSTARD [1] is a key inspiration for our work. **Our approach builds upon it by addressing two primary challenges, and our solutions to them represent the core innovations of our work.**
>
> 1. **Data Sourcing:** MUSTARD sources concepts from Khan Academy Website, which can create a "formalization bottleneck" as some concepts lack a direct counterpart in Mathlib. To avoid this, ATLAS sources concepts exclusively from Mathlib, guaranteeing a valid formalization path for every statement from the start.
>
> 2. **Generation Efficiency:** MUSTARD's reliance on costly GPT-4 correction loops is a high-cost, low-yield process, with over 90% of initial generations failing prover validation and ultimately producing only ~6k samples. In contrast, ATLAS's teacher-student distillation framework is highly efficient, dramatically lowering costs and enabling us to generate our much larger dataset of 117k high-quality pairs.
>
> **RAutoformalizer:** RAutoformalizer [2] is a dependency-based retrieval approach. This is complementary to our work. **The key distinction is that ATLAS focuses on generating a high-quality training dataset from scratch, while RAutoformalizer improves performance by retrieving relevant premises at inference time.**
>
> - A promising future direction is to extend our work from formalizing individual statements to mathematical theories. We propose achieving this by integrating our ATLAS data generation framework with dependency-aware methods like RAutoformalizer.
>
> -  A concrete technical path involves restructuring our mathematical concept repository into a dependency-aware tree. During synthesis, we would generate data in a bottom-up, layer-by-layer fashion, explicitly preserving the logical dependencies between foundational axioms and more advanced theorems. The resulting structured dataset would be ideal for training next-generation autoformalization models, paving the way for true theory-level autoformalization.
>
>
> We hope these clarifications and additions have fully addressed your concerns, and we are grateful for the opportunity to improve our paper based on your feedback.
>
> ---
>
> [1] Huang, Yinya, et al. "MUSTARD: Mastering Uniform Synthesis of Theorem and Proof Data." The Twelfth International Conference on Learning Representations. 2024.
>
> [2] Liu, Qi, et al. "Rethinking and improving autoformalization: towards a faithful metric and a dependency retrieval-based approach." The Thirteenth International Conference on Learning Representations. 2025.

---

### Note · Authors · 2025-08-12

Dear AC and Reviewers,

We sincerely thank you for your thoughtful evaluations, constructive questions, and recognition of our work. We are grateful for the positive feedback on our scalable, human-free framework for generating high-quality data and its resulting strong performance on an affordable budget. The constructive comments focused on expanding our evaluation and adding further validation, which we have now completed.

In direct response to this feedback, we performed extensive new experiments. By using full-parameter fine-tuning on multiple base models, we have established a new, decisive SOTA. For instance, our ATLAS Translator now surpasses the HERALD baseline by +23 absolute points on ProofNet (pass@1), more than doubles its performance on PutnamBench, and shows comparable results on miniF2F.

To enhance rigor, as suggested by reviewers, we:
* Adopted the stricter **BEq@k evaluation metric**.
* Conducted a new study on the **mathematical correctness** of our synthetic data.
* Added a new **out-of-domain evaluation** on the Con-NF benchmark.
* Quantitatively analyzed our **data augmentation diversity**.

These additions and new SOTA results will be fully integrated into the final manuscript. We are grateful that all reviewers responded positively to these clarifications—either increasing their scores or maintaining their support for acceptance. We appreciate the reviewers’ and AC’s time and consideration, and we hope our responses help finalize a favorable decision.

Sincerely,

The Authors

---

### Decision · Program_Chairs · 2025-09-17

**Decision:**

Accept (poster)

**Comment:**

This paper gives treatment to autoformalization. The authors responded very well to the concerns by the reviewers, and the paper now I believe is ready to be published. The author also did significant additional work during the rebuttal to improve the paper.